# Non-Stationary Contextual Pricing with Safety Constraints

**Dheeraj Baby** [*]                                                    *dheeraj@cs.ucsb.edu*
*Department of Computer Science*
*University of California, Santa Barbara*

**Jianyu Xu** [*]                                                       *xu_jy15@cs.ucsb.edu*
*Department of Computer Science*
*University of California, Santa Barbara*

**Yu-Xiang Wang**                                                       *yuxiangw@cs.ucsb.edu*
*Department of Computer Science*
*University of California, Santa Barbara*

**Reviewed on OpenReview:** *https://openreview.net/forum?id=fWIQ9Oaao0*

## Abstract

In a contextual pricing problem, a seller aims at maximizing the revenue over a sequence of sales sessions (described by feature vectors) using binary-censored feedback of "sold" or "not sold". Existing methods often overlook two practical challenges (1) the best pricing strategy could change over time; (2) the prices and pricing policies must conform to hard constraints due to safety, ethical or legal restrictions. We address both challenges by solving a more general problem of *universal dynamic regret* minimization in *proper* online learning with exp-concave losses — an open problem posed by Baby & Wang (2021) that we partially resolve in this paper, with attention restricted to loss functions coming from a generalized linear model. Here "dynamic regret" measures the performance relative to a non-stationary sequence of policies, and "proper" means that the learner must choose *feasible* strategies within a pre-defined convex set, which we use to model the safety constraints. In this work, we consider a linear noisy valuation model for the customers. With the market noises drawn from a *known strictly log-concave* distribution, our algorithm achieves $\tilde{O}(d^3 T^{1/3} C_T^{2/3} \vee d^3)$ dynamic regret in comparison with the optimal policy series, where $T$, $d$ and $C_T$ stand for the time horizon, the feature dimension and the total variation (characterizing non-stationarity) respectively. This regret is near-optimal with respect to $T$ (within $O(\log T)$ gaps) and $C_T$, and our algorithm is adaptable to *unknown $C_T$* and remains *feasible* throughout. However, the dependence on $d$ is suboptimal and the minimax rate is still open.

## 1 Introduction

Feature-based dynamic pricing, or *contextual* pricing, is a problem where the seller sets prices for different products based on their features and aims to maximize revenue. In general, a customer will make her decision based on a comparison between the price and her own valuation of the product. Formally, many existing works (e.g., Cohen et al., 2020; Javanmard & Nazerzadeh, 2019; Xu & Wang, 2021; Luo et al., 2021) adopt the following linear-feature valuation model:

---

[*]for equal contribution.

---

**Contextual pricing.** For $t = 1, 2, ..., T$:

1. A context $x_t \in \mathbb{R}^d$ is revealed that describes a sales session (product, customer and context).
2. The customer valuates the product as $y_t = x_t^\top \theta_t^* + N_t$ using $x_t$.
3. The seller proposes a price $v_t > 0$ concurrently (according to $x_t$ and historical sales records).
4. The transaction is successful if $v_t \leq y_t$, i.e., the seller gets a reward (payment) of $r_t = v_t \cdot \mathbb{1}(v_t \leq y_t)$.

---

Here $T$ is the unknown time horizon, $x_t$'s are adversarial features (which can be stochastic or non-stochastic series), $\theta_t^*$'s are *hidden parameters* mapping features to valuations linearly, and $N_t$'s are i.i.d. noises drawn from a known distribution $\mathbb{D}$. Denote $\mathbb{1}_t := \mathbb{1}(v_t \leq y_t)$ as the *Boolean-censored* feedback that equals 1 if $v_t \leq y_t$ and 0 otherwise, and we only observe $\mathbb{1}_t$ instead of the realized $y_t$ at each round. Our goal is to maximize the cumulative expected reward, and the *regret* is defined as the difference of expected rewards between $v_t$ and the best price at each round.

**Time-variant Behavior and Dynamic Regret.** Comparing with existing linear contextual pricing problem settings (e.g., Cohen et al., 2020; Javanmard & Nazerzadeh, 2019; Xu & Wang, 2021) where the linear valuation parameter $\theta_t^*$ is fixed as the same $\theta^*$ over all $t$, in this work we allow moderate changing of customers' valuations: i.e. $\theta_t^*$'s can vary over time, and the *total variation* $\sum_{t=1}^{T-1} \|\theta_t^* - \theta_{t+1}^*\|_1$ is upper bounded by some $C_T$ (which could be unknown to the seller). Here we adopt the $L_1$-norm bound because it is a reasonable metric for capturing the non-stationarity of the valuation mechanism: For instance, suppose each element of $x_t$ indicates the amount of one component of this product, and therefore each element of $\theta_t^*$ indicates the unit price of this component. In this example, $\|\theta_t^* - \theta_{t+1}^*\|_1$ reflects the general price fluctuations on the market, i.e., the sum of market-wise price changes over all components. To characterize the performance of a pricing scheme under this non-stationary setting, we adopt the concept of *dynamic regret*. In this notion, we compare the performance of $v_t$ we propose with that of the optimal pricing policy that knows the sequence of $\theta_t^*$ in advance. A rigorous definition of this dynamic regret will be presented in Section 2.3.

**Proper Learning.** Usually, the actions/strategies we are allowed to adopt are restricted in some specific *safe domains*. Taking any action/strategy outside this domain would probably cause risky, illegal or inconsistent outcomes. Our algorithm works by maintaining an estimate, $\theta_t$, for the true valuation parameter $\theta_t^*$ at each round $t$, and we in turn take $\theta_t$ as a *parametric strategy* for proposing the price $v_t$ according to a greedy policy (see Section 2.3 for more details). In this work, we require that the estimate $\theta_t$ must fall in a specific convex and closed domain $\mathcal{D}_t$ at each round $t$. Here $\mathcal{D}_t$ can be chosen adversarially with the constraint imposed by Assumption 1. As will be explained in Section 2.4, this is to address the fact that pricing strategies must conform to hard constraints due to safety restrictions.

**Universal Dynamic Regret and Proper OCO with co-variates.** Next, we take a digression and describe a general Online Convex Optimization (OCO) setting which will play a pivotal role in solving the contextual pricing problem.

---

**Proper OCO with co-variates.** For $t = 1, 2, ..., T$:

1. Adversary reveals a co-variate $x_t \in \mathbb{R}^d$.
2. Learner makes a decision $\hat{\theta}_t$ in a convex domain $\mathcal{D}_t \subset \mathbb{R}^d$.
3. Adversary reveals a convex loss function $\ell_t(\theta) = g_t(\theta^T x_t)$.

---

This setting embodies OCO under a wide range of loss functions from the generalized linear model (GLM) family for appropriate choices of $g_t$. The co-variates $x_t$ can be thought of as a feature that encodes valuable information about the context in round $t$ which can be used by the learner to make its predictions. Examples of this setting include (but are not limited to) linear regression and logistic regression.

The goal of the learner is to control its universal dynamic regret:

$$R(w_{1:T}) := \sum_{t=1}^{T} \ell_t(\hat{\theta}_t) - \ell_t(w_t), \tag{1}$$

where $w_{1:T} = \{w_1, w_2, \ldots, w_T\}$ is *any* comparator sequence satisfying $w_t \in \mathcal{D}_t$ for all $t \in [T]$. This is known to be a good metric in characterizing the performance of a learner in non-stationary environments (Zinkevich, 2003). Dynamic regret bounds are usually expressed in literature as functions of the time horizon $T$ and a path length that captures the smoothness of the comparator sequence such as $C_T = \sum_{t=1}^{T-1} \|w_t - w_{t+1}\|_1$.

## 1.1 Summary of Contributions

Our main contributions are given below.

1. We present an algorithm ProDR (Algorithm 1) that attains an *optimal* $\tilde{O}(d^3(T^{\frac{1}{3}} C_T^{\frac{2}{3}} \vee 1))$ dynamic regret (modulo dependencies in $d$ and $\log T$) for the setting of proper OCO with co-variates under exp-concave losses (see Section 3.1).

2. We construct an algorithm PDRP (Algorithm 2) with a base learner ProDR, which solves the non-stationary contextual pricing problem with strictly log-concave noise. We define the dynamic regret of contextual pricing as Eq.(3) and show that PDRP achieves a $\tilde{O}(d^3(T^{\frac{1}{3}} C_T^{\frac{2}{3}} \vee 1))$ dynamic regret guarantee (see Section 3.2).

3. We show that any algorithm must incur a dynamic regret of $\Omega(T^{\frac{1}{3}} C_T^{\frac{2}{3}} \vee 1)$ in the contextual pricing problem, which says that PDRP is minimax optimal up to $d$ and $\log T$ factors (see Section 3.3).

**Novelty.** Owing to the reduction of Xu & Wang (2021), the non-stationary contextual pricing problem can be reduced to an OCO problem with co-variates and exp-concave losses. The key subroutine we developed — ProDR — is the first to achieve an *optimal* universal dynamic regret with *exp-concave* losses in the *proper OCO with covariate* setting. ProDR makes considerable progress towards addressing the open problem posed by Baby & Wang (2021) on the more general version of the above problem with *general exp-concave* losses (rather than GLM with known covariates) . The only existing attempt to this open problem requires the decision set to be an $L_\infty$ ball (Baby & Wang, 2022b), which cannot be used to handle arbitrary convex decision sets as we do.

**Summary of techniques.** The key technique in deriving ProDR is a novel "transfer theorem" which takes the algorithm of Baby & Wang (2022b) ($L_\infty$ ball decision set) and converts it to an optimal algorithm for the setting of proper OCO with co-variates under *arbitrary convex decision sets*. This idea is similar in spirit to the improper-proper reduction in the work of Cutkosky & Orabona (2018) where they consider general convex losses. However, a direct application of their reduction scheme cannot give fast rates for exp-concave losses. To circumvent this issue, we propose new reduction schemes that carefully take the curvature of the losses into account thereby allowing us to derive fast and optimal dynamic regret rates under exp-concave losses (see Section 3.1 for a list of technical challenges). Such a "transfer theorem" could be of independent interest and impactful in the general context of non-stationary online learning. That the non-stationary dynamic pricing problem can be optimally solved using ProDR is a testament to this fact.

## 1.2 Related Works

Here we discuss how our work relates to the existing literature on dynamic pricing and dynamic regret.

**Dynamic Pricing** Dynamic pricing has been extensively studied under the single product (non-contextual) setting (e.g. Kleinberg & Leighton, 2003; Besbes & Zeevi, 2015; Wang et al., 2021), where the goal is to find and approach the best fixed price that maximizes the expected revenue. The problem is later generalized to *contextual pricing* where a feature $x_t$ occurs at each time $t$ and the customer's valuation is dependent on $x_t$. A widely adopted model is the *linear valuation* (e.g., Cohen et al., 2020; Javanmard & Nazerzadeh, 2019; Xu & Wang, 2021), where they assume all customers' valuations are a fixed feature-to-valuation mapping (i.e., $\theta_t^*$ is fixed, $\forall t$) adding i.i.d. noises drawn from a known distribution. As a result, the best price varies on different features occurring over time, and the goal turns to approach the best price in every round. However, the optimal pricing policy is static and the regret definition is a comparison of performance between our proposed price and the optimal policy that knows $\theta^*$ and the noise distribution in advance. In this work, we adopt this linear valuation setting and further generalize to non-stationary cases where the linear mapping $\theta_t^*$ is changeable over time. As a result, the best pricing policy also changes according to $\theta_t^*$, and we have to analyze the algorithmic performance in the scale of *dynamic regret*. Leme et al. (2021) also studies non-stationary

pricing problems and adopts the dynamic regret metric. However, their loss function and constraints are defined on *realized* valuations, different from our regret function defined on the *expected* valuations. Also, they adopt a non-contextual problem setting and assume individual variation bounds on each pair of adjacent valuations like $|v_{t+1} - v_t| \leq \epsilon_t$ for some $\epsilon_t > 0, t = 1, 2, \ldots, T$. Under their assumptions, they achieve an $\tilde{O}(T^{\frac{1}{2}} C_T^{\frac{1}{2}})$ optimal regret with a matching lower bound[1]. Given those differences in problem settings, their regret bounds are not applicable to our problem.

Some existing works on stationary pricing problems adopt contextual bandits to achieve sub-linear static regrets (Luo et al., 2021; Xu & Wang, 2022). Therefore, it is also possible to reduce our non-stationary pricing problem to a non-stationary bandit problem (Chen et al., 2019; Cheung et al., 2022; Zhao et al., 2020a). However, similarly to what Amin et al. (2014) indicates in stationary contextual pricing, a direct application of non-stationary bandit algorithms to our known-noise-distribution setting might cause sub-optimality in dynamic regret. Such a reduction might be optimal without the knowledge of noise distribution, but it is beyond the scope of this work.

Besides, a recent stream of works study the pricing problems under constraints on inventory (Chen & Gallego, 2022), reserve-price (Niu et al., 2020), fairness (Cohen et al., 2021; 2022; Xu et al., 2023; Chen et al., 2023) and budget (Salehi & Mirmohammadi, 2023), etc. Our work also contributes to this series as we enforce safety constraints and adaptively measure the impact of those constraints on the dynamic regret.

**Dynamic Regret.** There is a rich body of literature aimed in minimizing the universal dynamic regret (Eq.(1)) in OCO setting where the earliest works can be traced back to Zinkevich (2003). When the revealed losses are convex, Zhang et al. (2018) proposes algorithms to attain an optimal dynamic regret rate of $O(\sqrt{T(1 + P_T)})$ where $P_T = \sum_{t=1}^{T-1} \|w_t - w_{t+1}\|_2$. When the loss functions are gradient Lipschitz, problem dependent regret bounds have been developed in the work of Zhao et al. (2020b). In addition to gradient Lipschitzness, if the losses have extra curvature properties such as exp-concavity, Baby & Wang (2021) proposes algorithms that attain a near optimal dynamic regret of $\tilde{O}^*(T^{1/3} C_T^{2/3} \vee 1)$ ($\tilde{O}^*$ hides dependencies on dimensions and factors of $\log T$). The work of Baby & Wang (2022b) shows similar rates for non-smooth and exp-concave losses in a proper learning setting when the decision set is an $L_\infty$ ball. In contrast, our work is able to attain near optimal rates for arbitrary convex decision sets for a large family of exp-concave losses. Further Baby & Wang (2022b) also shows optimal rates for arbitrary bounded convex decision sets when the losses are strongly convex.

If we take all the comparators $w_t$ in Eq.(1) to be same, one recovers the notion of static regret. There are works that aim in controlling the static regret in any time window which makes them suitable for learning in non-stationary environments. These algorithms fall into the category of adaptive / strongly adaptive regret minimization algorithms. Examples of such methods include Hazan & Seshadhri (2007); Daniely et al. (2015); Adamskiy et al. (2016); Jun et al. (2017); Cutkosky (2020); Baby et al. (2021); Zhang et al. (2021). We refer the readers to Baby & Wang (2021) and references therein for a more inclusive survey on dynamic regret and strongly adaptive algorithms.

There has also been recent advances (e.g. Zhao et al., 2022; Baby & Wang, 2022a) in applying online learning techniques to design controllers. In particular, Baby & Wang (2022a) uses a reduction from Linear Quadratic Regulator (LQR) problem to online (mini-batch) linear regression problem due to the work of Foster & Simchowitz (2020). They employ a black-box reduction technique to convert the algorithm of Baby & Wang (2022b) to one that attains optimal dynamic regret for online linear regression under the setting of proper learning. This is facilitated by exploiting the constant hessian property of linear regression losses. However, this property will not be satisfied for exp-concave losses in general. As such it is unclear that the black-box reduction techniques of Baby & Wang (2022a) are generalisable beyond linear regression. Hence the results in this paper are not directly implied by their results. We direct the readers to Baby & Wang (2022b) for an elaborate discussion about the literature on online learning and control.

---

[1]Here we reduce their loss bound to our notations.

## 2 Notations and Problem Setup

In this section, we specify necessary mathematical symbols and notations, and define functions for algorithm design and regret analysis. We also present three examples to illustrate the concept of proper learning in contextual pricing.

### 2.1 Symbols and Notations.

The pricing process consists of $T$ rounds. $x_t, \theta_t^* \in \mathbb{R}^d$, $y_t \in \mathbb{R}$, $v_t \in \mathbb{R}_+$ and $N_t \in \mathbb{R}$ denote the feature vector, the linear valuation parameter, the customer's valuation, the seller's price and the noise at time $t$, sequentially. At each round, we receive a payoff (reward) $r_t = v_t \cdot \mathbb{1}_t$, where the binary variable $\mathbb{1}_t$ indicates the *customer's decision*, i.e., $\mathbb{1}_t = \mathbf{1}(v_t \leq y_t)$.

### 2.2 Technical Assumptions

Denote a norm-bounded domain family $\mathcal{D}_p^B = \{\theta \in \mathbb{R}^d, \|\theta\|_p \leq B\}$. We firstly present assumptions on domain constraints of $x_t$ and $\theta_t^*$:

**Assumption 1** (Domain Constraints). Assume $x_t \in \mathcal{D}_x$ where $\mathcal{D}_x \subseteq \mathcal{D}_2^1$ is convex and closed, and $\theta_t^* \in \mathcal{D}_t$ where every $\mathcal{D}_t \subseteq \mathcal{D}_2^B \subset \mathcal{D}_\infty^B$ is also convex and closed. Each $\mathcal{D}_t$ can be chosen adversarially and is known to the learner before time $t$.

Here we want the customers' valuations to be bounded. Equivalently, we may also assume that $\mathcal{D}_x \subseteq \mathcal{D}_2^{B_1}$ and $\mathcal{D}_t \subseteq \mathcal{D}_2^{B/B_1}$ for any $B_1 > 0$. With these assumptions, we know that $|x_t^\top \theta| \leq B, \forall \theta \in \mathcal{D}_2^B, t = 1, 2, \ldots, T$. Next, we make a reasonable assumption on customers' expected valuations:

**Assumption 2** (Non-Negative Expected Valuation). For a customer's valuation $y_t = x_t^\top \theta_t^*$, we assume the *expected valuation* $x_t^\top \theta_t^* \geq 0, t = 1, 2, \ldots, T$.

Now we make assumptions on the distribution of noise $N_t$. We firstly present the definitions of *log-concavity* and *strict log-concavity* on 1-dimensional distributions according to Prékopa (1973).

**Definition 2.1** (Log-concavity and strict log-concavity). A probability measure $P$ defined on $\mathbb{R}$ is said to be *log-concave* if and only if for any pair $A, B \subset \mathbb{R}$ of intervals, it holds that

$$P(\lambda A + (1 - \lambda)B) \geq \{P(A)\}^\lambda \{P(B)\}^{1-\lambda}, \forall \lambda \in (0, 1).$$

Here "+" denotes Minkowski addition. Also, $P$ is *strictly log-concave* if and only if for any pair $A, B \subset \mathbb{R}$ of intervals, $A \neq B$, it holds that

$$P(\lambda A + (1 - \lambda)B) > \{P(A)\}^\lambda \{P(B)\}^{1-\lambda}, \forall \lambda \in (0, 1).$$

Then we make the following assumption:

**Assumption 3** (Valuation noise distribution). At each time $t = 1, 2, \ldots, T$, the noise $N_t$ is independently and identically sampled from a fixed *strictly log-concave* distribution $\mathbb{D}$ with a twice continuously differentiable cumulative distribution function (CDF) $F$. Furthermore, the first and second derivatives of the CDF, denoted as $f$ and $f'$, respectively are bounded by two finite constants $B_f := \sup_{\omega \in \mathbb{R}} f(\omega)$ and $B_{f'} := \sup_{\omega \in \mathbb{R}} |f'(\omega)|$.

According to Definition 2.1, let (i) $A = (-\infty, x], B = (-\infty, y]$ and (ii) $A = (x, +\infty), B = (y, +\infty)$ respectively, and we have $F$ and $(1 - F)$ are both strictly log-concave functions. Existing works on contextual pricing also adopt log-concavity assumptions (see Javanmard & Nazerzadeh, 2019). For a detailed discussion on log-concave distributions, we kindly refer the audience to Bagnoli & Bergstrom (2006).

All of those three assumptions are supposed to hold throughout the paper.

### 2.3 Functions and Key Quantities

**Greedily Pricing.** Here we adopt two functions defined by Xu & Wang (2021) and also make use of their properties. Firstly, we introduce an *expected reward* function $g(v, u) := \mathbb{E}[r_t | v_t = v, x_t^\top \theta^* = u] = v \cdot (1 - F(v - u))$ that is unimodal w.r.t. $v$. Secondly, we introduce a *greedily pricing* function $J(u) := \operatorname{argmax}_{v \in \mathbb{R}} g(v, u)$. $J(u)$ has two important properties: On the one hand, $J(u)$ is strictly monotonically

increasing, with $J'(u) \in (0,1)$. Therefore, $J(u)$ and $J^{-1}(v)$ are bijections, $\forall u \in \mathbb{R}, v > 0$. On the other hand, we have $\|\nabla_\theta J(x^\top \theta)\|_2 = |J'(x^\top \theta)| \cdot \|x\|_2 \leq 1$, which guarantees a low price-changing rate while modifying parameter $\theta$.

**Restrictions on Actions/Parametric Strategies.** When we take an action by presenting a price $v_t$, there always exists an $\theta_t \in \mathbb{R}^d$ such that $x_t^\top \theta_t = J^{-1}(v_t)$. Therefore, for any price $v_t > 0$, it is equivalent to firstly propose a corresponding *parametric strategy* $\theta_t$ (satisfying $x_t^\top \theta_t = J^{-1}(v_t)$) and then set the price as $J(x_t^\top \theta_t)$. Since we are approaching the optimal price (which is $J(x_t^\top \theta^*)$) and that $\theta_t^* \in \mathcal{D}_t$, we may restrict the strategy $\theta_t$ to be taken within $\mathcal{D}_t$ at each time $t$. We will explain more on the motivation of the restrictions in Section 2.4.

**Negative Log-likelihood.** We define

$$\ell_t(\theta) = -\mathbb{1}_t \cdot \log\left(1 - F(v_t - x_t^\top \theta)\right) - (1 - \mathbb{1}_t) \log\left(F(v_t - x_t^\top \theta)\right) \tag{2}$$

as a negative log-likelihood function at round $t$. Also, we define an expected log-likelihood function $L_t := \mathbb{E}_{N_t}[\ell_t(\theta)|x_t, \theta_t^*]$. For the simplicity of notations in the following sections, we denote $h_t(\theta) := \frac{\partial \ell_t(\theta)}{\partial x_t^\top \theta} \in \mathbb{R}$, and we show a property of $h_t(\theta)$:

**Lemma 2.2.** *For $\theta \in \mathcal{D}_2^B$, there exist constants $0 < h_{\min} \leq h_{\max} < +\infty$ such that*

$$h_{\max} = \sup_{\theta \in \mathcal{D}_2^B} |h_t(\theta)|, h_{\min} = \inf_{\theta \in \mathcal{D}_2^B} |h_t(\theta)|, \forall t = 1, 2, \ldots, T.$$

We prove this by noticing that $h(\theta)$ is continuous and $\mathcal{D}_2^B$ is closed, and the details are in Appendix B.1. With this lemma, we may know that $\ell_t(\theta)$ is Lipschitz (see Lemma 3.8).

**Dynamic Regret.** Finally, we define the cumulative *dynamic regret*:

$$\mathbf{Reg}_T = \sum_{t=1}^{T} g(J(x_t^\top \theta_t^*), x_t^\top \theta_t^*) - g(v_t, x_t^\top \theta_t^*). \tag{3}$$

We usually measure the regret as a function of $T, d$ and the total variation $C_T := \sum_{t=1}^{T-1} \|\theta_t^* - \theta_{t+1}^*\|_1$.

### 2.4 Examples

Here we present three examples where the nature requires the strategies to lie in a "safe domain", regarding risk-taking, legal or consistency concerns.

**Risk Control** Adopting strategies outside a pre-defined and protected decision set can be very risky in general. Concerning our contextual pricing problem, an extremely low price would lead to significant loss of profit. Therefore, we have to set a lower pricing bar for each item. At each time $t$, suppose the lower bar is $c_t > 0$, and therefore our parametric strategy $\theta_t$ should satisfy $c_t \leq J(x_t^\top \theta_t)$. Since $J(u)$ is monotonically increasing, we have $x_t^\top \theta_t \geq J^{-1}(c_t)$. By intersecting $\{\theta \in \mathbb{R}^d | x_t^\top \theta \geq J^{-1}(c_t)\}$ with the $L_2$-norm ball $\mathcal{D}_2^B$, we get a convex and compact set $\mathcal{D}_t$, in which any parametric strategy $\theta$ satisfies the norm bound and will lead to a price not less than $c_t$ given the $J(x_t^\top \theta)$ greedy pricing policy.

**Legal Concern** There exist laws or regulations regarding the highest price of some specific products. For each item with feature $x_t$, suppose that we cannot set a price exceeding $c_t > 0$. Equivalently, the parametric strategy $\theta_t$ we take must satisfy $v_t = J(x_t^\top \theta_t) \leq c_t$. Since $J(u)$ is monotonically increasing, this is further equivalent to $x_t^\top \theta_t \leq J^{-1}(c_t)$. Therefore, the restricted strategy space $\mathcal{D}_t$ is the intersection of $\{\theta | x_t^\top \theta \leq J^{-1}(c_t)\}$ with the $L_2$-norm ball $\mathcal{D}_2^B$, which is a convex and compact set. Any parametric strategy falling out of $\mathcal{D}_t$ would lead to either $v_t > c_t$ or $\|\theta\| > B$.

**Price Consistency** It is important for the seller to be consistent on setting prices, or otherwise it might cause pricing discrimination. Specifically, if two identical items with feature $x$ occur at time $t$ and $t + 1$, then their prices must be close to each other. In other words, we require $|J(x^\top \theta_t) - J(x^\top \theta_{t+1})| \leq C, \forall x \in \mathcal{D}_x \subset \mathcal{D}_2^1$ for some constant $C > 0$. For each $x \in \mathcal{D}_x$, we may solve it and get

---

**Algorithm 1** Proper Dynamic Regret minimization (ProDR)

---

1: **Input:** Base algorithm $\mathcal{A}$, barrier multiplier $G' > 0$, exp-concavity factor $\beta$.
2: **for** $t = 1, 2, \ldots, T$: **do**
3:    Get iterate $\tilde{\theta}_t$ from $\mathcal{A}$.
4:    Feature $x_t$ and proper domain $\mathcal{D}_t$ are revealed
5:    Output $\hat{\theta}_t = \arg\min_{\theta \in \mathcal{D}_t} |x_t^\top (\theta - \tilde{\theta}_t)|$.
6:    Loss $\ell_t$ is revealed.
7:    Construct $\hat{\ell}_t(\theta)$ as in Eq.(4) and set

$$f_t(\theta) = \hat{\ell}_t(\theta) + G' \cdot S_t(\theta),$$

   where $S_t(\theta) = \min_{\eta \in \mathcal{D}_t} |\nabla \hat{\ell}_t(\hat{\theta}_t)^\top (\eta - \theta)|$;
8:    Send $f_t(\theta)$ to $\mathcal{A}$ as loss at time $t$.
9: **end for**

---

$$J^{-1}(J(x^\top \theta_t) - C) \leq x^\top \theta_{t+1} \leq J^{-1}(C + J(x^\top \theta_t)).$$

Denote this set as $\mathcal{S}_t(x)$, and we have $\mathcal{D}_{t+1} \subseteq \cap_{x \in \mathcal{D}_x} \mathcal{S}_t(x)$. Since $\theta_t \in \mathcal{S}_t(x), \forall x$, the intersection is non-empty.

## 3   Main Results

In this section, we present and analyse our algorithms. In Section 3.1, we first study the more general problem of universal dynamic regret (Eq.(1)) minimization in a proper OCO setting. Results of Section 3.1 will be applied in Section 3.2 to derive an optimal algorithm for the non-stationary pricing problem. All omitted proofs in this section are deferred to Appendix B.

### 3.1   Dynamic Regret of ProDR

In this section, we study the **Pro**per **D**ynamic **R**egret minimization (ProDR) algorithm (Algorithm 1). We consider the protocol of proper OCO with co-variates introduced in Section 1.

The goal of this section is to control the universal dynamic regret as defined in Eq.(1). We start by listing out the assumptions we made for the OCO problem.

**Assumption 4.** A constant $B > 0$ is known such that $\max_{\theta \in \mathcal{D}_t} \|\theta\|_\infty \leq B$ for all $t \in [n]$.

**Assumption 5.** The losses $\ell_t$ obey $\|\nabla \ell_t(\theta)\|_2 \leq G$ for all $t \in [n]$ and $\theta \in \mathcal{D}_t$ (recall that $\mathcal{D}_t \subseteq \mathcal{D}_2^B$ from Section 2.2).

**Assumption 6.** The losses are $\alpha$ exp-concave. i.e $\ell_t(y) \geq \ell_t(x) + \nabla \ell_t(x)^\top (y - x) + \frac{\alpha}{2} \left( \ell_t(x)^\top (y - x) \right)^2$, for $\alpha > 0$ and for all $x, y \in \mathcal{D}_2^B$.

Assumption 4 puts a relatively mild constraint that a box enclosing all the decision sets is known ahead of time. Lipschitzness assumptions like Assumption 5 are standard in online learning. Assumption 6 states that the loss $\ell_t$ exhibits a strong curvature in the direction of its gradients (see Hazan et al., 2007, as an example). We will exploit this curvature to derive fast regret rates.

**Qualitative description of ProDR.** The base algorithm $\mathcal{A}$ in ProDR is expected to optimally control the dynamic regret under exp-concave losses and when the decision set is a box: $\mathcal{D}_\infty^B = \{x \in \mathbb{R}^d : \|x\|_\infty \leq B\}$, where $B$ is as in Assumption 4. The idea is to perform a black-box reduction that can convert the base algorithm $\mathcal{A}$ to an algorithm that attains good dynamic regret guarantee on the domains $\mathcal{D}_t$. Though similar ideas have been already explored in the work of Cutkosky & Orabona (2018), our way of constructing such reductions for the current problem is new and interesting in its own right in the context of exp-concave online learning. Next, we expand upon this matter highlighting the differences from Cutkosky & Orabona (2018). We construct losses $f_t$ in Line 7 of ProDR where the $S_t(\theta)$ term acts as a regularizer that penalizes $\mathcal{A}$ for

predicting points outside $\mathcal{D}_t$. We would like the losses $f_t$ to be exp-concave as the base algorithm $\mathcal{A}$ expects. However, a direct application of the techniques of Cutkosky & Orabona (2018) does not satisfy this property. We address this issue by carefully constructing $f_t$ as in Line 7 of Algorithm 1 such that: 1) gradients of both $\hat{\ell}_t(\theta)$ and $S_t(\theta)$ lie in the span of co-variate $x_t$ and 2) $\hat{\ell}_t(\theta)$ is exp-concave, meaning that it exhibits strong curvature along the direction of $x_t$. Now, 1 and 2 together implies that the surrogate losses $f_t$ still remains exp-concave as it exhibits strong curvature along the direction of its gradient (which is spanned by $x_t$). The particular choice of $\hat{\ell}_t(\theta)$ is found to be crucial in preventing the exp-concavity factor of losses $f_t$ from collapsing to zero. We will show that the dynamic regret of ProDR w.r.t. losses $\hat{\ell}_t$ is upper bounded by the dynamic regret of the base algorithm $\mathcal{A}$ wrt losses $f_t$ which is well controlled.

We next describe the dynamic regret guarantees of Algorithm 1. We inherit all the notations used in the algorithm description.

**Theorem 3.1.** *Let $\beta = \min\{\alpha/2, 1/(8GB\sqrt{d})\}$ and $\gamma = \frac{1}{4\left(2GB\sqrt{d\beta}+1/(2\sqrt{\beta})\right)^2}$ and $G' = 1 + 2GB\beta\sqrt{d}$. Let $\mathcal{A}$ in ProDR algorithm be FLH-ONS (Fig.1 in Appendix A) instantiated with parameters $\zeta = 2\gamma/25$, $\mathcal{G} = GG'$ and $\phi = B$. Then ProDR (Algorithm 1) satisfes*

$$\sum_{t=1}^{T} \ell_t(\hat{\theta}_t) - \ell_t(w_t) = \tilde{O}\left((d^3\gamma + \frac{d^2}{\gamma})(T^{1/3}C_T^{2/3} \vee 1)\right),$$

*where $C_T := \sum_{t=2}^{T} \|w_t - w_{t-1}\|_1$ with $w_t \in \mathcal{D}_t$. $a \vee b := \max\{a, b\}$ and $\tilde{O}$ hides dependence of constants $G, B, \alpha$ and poly-logarithmic factors of $T$.*

*Remark* 3.2 (Adaptivity to $C_T$). In light of the $\Omega(dB^2 \log T \vee d^{1/3}T^{1/3}C_T^{2/3}B^{4/3})$ lower bound (see Baby & Wang, 2021, Proposition 11), we see that the ProDR algorithm adapts optimally to the path variation $C_T$ of the comparator sequence, which may not be known ahead of time.

*Proof.* Due to the $\alpha$ exp-concavity of losses $\ell_t$ over the domain $\mathcal{D}_2^B$ and $\beta \leq \frac{\alpha}{2}$ we have that:

$$\ell_t(\theta) \geq \ell_t(\hat{\theta}_t) + \nabla\ell_t(\hat{\theta}_t)^\top(\theta - \hat{\theta}_t) + \beta\left(\nabla\ell_t(\hat{\theta}_t))^\top(\theta - \hat{\theta}_t)\right)^2,$$

for any $\theta \in \mathcal{D}_2^B$. Hence following Hazan et al. (2007), we consider the linear-regression-type surrogate losses:

$$\hat{\ell}_t(\theta) := \left(\nabla\ell_t(\hat{\theta}_t)^\top(\theta - \hat{\theta}_t)\sqrt{\beta} + \frac{1}{2\sqrt{\beta}}\right)^2. \tag{4}$$

Hence for any $\theta \in \mathcal{D}_2^B$ we have that

$$\ell_t(\hat{\theta}_t) - \ell_t(\theta) \leq \frac{1}{4\beta} - \hat{\ell}_t(\theta) = \hat{\ell}_t(\hat{\theta}_t) - \hat{\ell}_t(\theta). \tag{5}$$

where we used the fact that $\hat{\ell}_t(\hat{\theta}_t) = \frac{1}{4\beta}$.

Given that $S_t(\theta_t^*) = S_t(\hat{\theta}_t) = 0$ since $\theta_t^*, \hat{\theta}_t \in \mathcal{D}_t$, we have

$$f_t(\theta_t^*) = \hat{\ell}_t(\theta_t^*), f_t(\hat{\theta}_t) = \hat{\ell}_t(\hat{\theta}_t). \tag{6}$$

Let us denote $\nabla \ell_t(\theta) = h_t(\theta) x_t$ where $h_t(\theta) = g_t'(x_t^\top \theta)$. Now, according to the definition of $S_t(\theta)$ and $\hat{\theta}_t$, we have:

$$
\begin{aligned}
f_t(\tilde{\theta}_t) =& \hat{\ell}_t(\tilde{\theta}_t) + G' \cdot S_t(\tilde{\theta}_t) \\
=& \hat{\ell}_t(\tilde{\theta}_t) + G' \cdot \min_{\eta \in \mathcal{D}_t} |\nabla \ell_t(\hat{\theta}_t)^\top (\eta - \tilde{\theta}_t)| \\
=& \hat{\ell}_t(\tilde{\theta}_t) + G' \cdot \min_{\eta \in \mathcal{D}_t} |h_t(\hat{\theta}_t)||x_t^\top (\eta - \tilde{\theta}_t)| \\
=& \hat{\ell}_t(\tilde{\theta}_t) + G' \cdot |h_t(\hat{\theta}_t)||x_t^\top (\hat{\theta}_t - \tilde{\theta}_t)| \\
=& \hat{\ell}_t(\tilde{\theta}_t) + G' \cdot |\nabla \ell_t(\hat{\theta}_t)^\top (\hat{\theta}_t - \tilde{\theta}_t)|.
\end{aligned}
$$

Next we proceed to upper bound the regret w.r.t. losses $\hat{\ell}_t$ by the regret w.r.t. losses $f_t$. We need the following lemma.

**Lemma 3.3.** *Under the assumptions of Theorem 3.1, we have that*

$$
|\hat{\ell}_t(\theta) - \hat{\ell}_t(\hat{\theta}_t)| \leq G' |\nabla \ell_t(\hat{\theta}_t)^\top (\theta - \hat{\theta}_t)|,
$$

*for any $\theta \in \mathcal{D}_\infty^B$ where $G' := (1 + 2GB\beta\sqrt{d})$.*

The proof is shown in Appendix B.2. With this lemma, we have

$$
\hat{\ell}_t(\hat{\theta}_t) \leq \hat{\ell}_t(\tilde{\theta}_t) + G' \cdot |\nabla \ell_t(\hat{\theta}_t)^\top (\hat{\theta}_t - \tilde{\theta}_t)| = f_t(\tilde{\theta}_t).
$$

Combining the above inequality with Eq.(6) we obtain

$$
\hat{\ell}_t(\hat{\theta}_t) - \hat{\ell}_t(\theta_t^*) \leq f_t(\tilde{\theta}_t) - f_t(\theta_t^*).
$$

Now using Eq.(5) along with the previous relation yields that

$$
\sum_{t=1}^T \ell_t(\hat{\theta}_t) - \ell_t(\theta_t^*) \leq \sum_{t=1}^T f_t(\tilde{\theta}_t) - f_t(\theta_t^*).
$$

The following lemma specifies how to compute the sub-gradient of the regularizer term $S_t(\theta)$ in Line 7 of Algorithm 1. Further it highlights an important property that a sub-gradient of $S_t(\theta)$ lies in the span of covariate $x_t$ (recall that $\nabla \ell_t(\theta) = h(\theta) x_t$). This is also useful for proving the joint exp-concavity of the losses $f_t$.

**Lemma 3.4.** *The function $S_t(\theta)$ is convex across $\mathbb{R}^d$. Denote $\eta_t(\theta) := \operatorname{argmin}_\eta |x_t^\top(\eta - \theta)|$. When $\nabla \ell_t(\hat{\theta}_t)^\top(\eta_t(\theta) - \theta) \neq 0$, we have:*

$$
\nabla S_t(\theta) = \begin{cases} \nabla \ell_t(\hat{\theta}_t), & if \quad \nabla \ell_t(\hat{\theta}_t)^\top(\eta_t(\theta) - \theta) < 0 \\ -\nabla \ell_t(\hat{\theta}_t), & if \quad \nabla \ell_t(\hat{\theta}_t)^\top(\eta_t(\theta) - \theta) > 0. \end{cases}
$$

*When $\nabla \ell_t(\hat{\theta}_t)^\top(\eta_t(\theta) - \theta) = 0$, we have $\mathbf{0} \in \partial S_t(\theta)$.*

The proof of Lemma 3.4 is in Appendix B.3. In the next lemma, we show that the losses $f_t$ remain exp-concave with appropriate exp-concavity factor **bounded away** from zero. This is the key lemma that helps to control the regret of ProDR.

**Lemma 3.5.** *Define $\gamma := \frac{1}{4\left(2GB\sqrt{d\beta} + 1/(2\sqrt{\beta})\right)^2}$. We have that the surrogate losses $f_t$ are $2\gamma/25$ exp-concave and $2GG'$ Lipschitz in $L_2$ norm across $\mathcal{D}_\infty^B$.*

---

**Algorithm 2** Proper Dynamic Regret Pricing (PDRP)

---

1: **Input:** Noise distribution $\mathbb{D}$ (including its CDF $F$ and PDF $f$).
    ProDR algorithm $\mathcal{A}$ instantiated as in Theorem 3.1.
2: **for** $t = 1, 2, \ldots, T$: **do**
3:    Feature $x_t$ and proper domain $\mathcal{D}_t$ are revealed and sent to $\mathcal{A}$.
4:    Get $\hat{\theta}_t \in \mathcal{D}_t$ from $\mathcal{A}$.
5:    Seller proposes $v_t = J(x_t^\top \hat{\theta}_t)$ and receive $\mathbb{1}_t$.
6:    Send loss $\ell_t(\theta)$ defined in Eq.(2) to $\mathcal{A}$.
7: **end for**

---

As is stated earlier in this section, the intuition of this lemma comes from two facts: (1) both $\nabla \hat{\ell}_t(\theta)$ and $\nabla S_t(\theta)$ are in the span of $x_t$, and (2) $\hat{\ell}_t(\theta)$ is exp-concave. As a result, the strong curvature of $\hat{\ell}_t(\theta)$ along the $x_t$ direction "absorbs" the plain convexity of $S_t(\theta)$ and therefore guarantees the exp-concavity of $f_t(\theta)$. We defer the detailed proof to Appendix B.4. Hence from Baby & Wang (2022b, Theorem 10) , FLH-ONS algorithm (Fig.1 in Appendix A) run with parameters $\zeta = 2\gamma/25$, $\mathcal{G} = GG'$ and $\phi = B$ can be used to control

$$
\sum_{t=1}^{T} f_t(\tilde{\theta}_t) - f_t(\theta_t^*) = \tilde{O}\left( d^2(G^2(G')^2 B^2 \gamma d + G^2(G')^2 B^2 + \frac{1}{\gamma})(T^{1/3}C_T^{2/3} \vee 1)\right)
$$
$$
= \tilde{O}\left( d^3(T^{1/3}C_T^{2/3} \vee 1)\right),
$$

where the last line is got by plugging in the values of $\gamma$ and $G'$ and upper bounding further.  ∎

## 3.2 Dynamic regret of PDRP

In this section, we present our main algorithm for controlling the dynamic regret on contextual pricing problem, the **P**roper **D**ynamic **R**egret **P**ricing (PDRP) (Algorithm 2).

**Qualitative description of PDRP.** Xu & Wang (2021) observes that the pricing problem can be reduced to the setting of proper OCO with co-variates and exp-concave losses. This observation, armed with the ProDR algorithm, naturally lends itself to the algorithm PDRP for controlling dynamic regret of the pricing problem.

We are now ready to present regret guarantees for the non-stationary pricing problem.

**Theorem 3.6.** *Consider the linear noisy contextual pricing problem defined in Section 1. Assume that we know the noise distribution $\mathbb{D}$ exactly. By properly initializing $\beta, \gamma$ and $G'$ with pre-knowledge, PDRP (Algorithm 2) obeys $\mathbf{Reg}_T = \tilde{O}(d^3(T^{\frac{1}{3}}C_T^{\frac{2}{3}} \vee 1))$, where $\mathbf{Reg}_T$ is as defined in Eq.(3), $\tilde{O}$ hides poly-logarithmic factors of $T$ and $(a \vee b) = \max\{a, b\}$.*

*Proof.* We start with the lemmas that help us leverage the OCO framework of Section 3.1.

**Lemma 3.7.** *(Xu & Wang, 2021, Lemma 5 and 6) Under the assumptions in Theorem 3.6, for $\theta \in \mathcal{D}_2^B$, we have:*

$$
g(J(x_t^\top \theta_t^*), x_t^\top \theta_t^*) - g(J(x_t^\top \theta), x_t^\top \theta_t^*) \le \frac{2C}{C_{down}} \left(E[\ell_t(\theta) - \ell_t(\theta_t^*)]\right),
$$

*where $\ell_t$ is defined in Eq.(2), $C = 2B_f + (B + J(0))B_{f'}$ and*

$$
C_{down} := \inf_{\omega \in [-B, B+J(0)]} \min\left\{ \frac{d^2 \log(1 - F(\omega))}{d\omega^2}, \frac{d^2 \log(F(\omega))}{d\omega^2}\right\} > 0.
$$

So we have

$$
\mathbf{Reg}_T \le \frac{2C}{C_{down}} \mathbb{E}[\ell_t(\hat{\theta}_t) - \ell_t(\theta_t^*)]. \tag{7}
$$

Next, we record the curvature and smoothness properties of losses $\ell_t$.

**Lemma 3.8.** *Let $G = h_{\max}$ defined in Lemma 2.2. Under the assumptions in Theorem 3.6, for $\theta \in \mathcal{D}_t$, we have: (1) $\ell_t(\theta)$ is G-Lipschitz in $\|\cdot\|_2$ norm, and (2) $\ell_t(\theta)$ $\frac{C_{down}}{C_{exp}}$-exp-concave. Here $C_{exp} := \sup_{\omega \in [-B, B+J(0)]} \max\left\{ \frac{f(\omega)^2}{F(\omega)^2}, \frac{f(\omega)^2}{(1-F(\omega))^2} \right\}$ and $C_{down}$ is defined in Lemma 3.7 .*

This lemma is derived from Xu & Wang (2021) Lemma 7, and we defer the proof to Appendix B.5. The lemma above implies that the losses satisfy Assumption 5 in Section 3.1. Further they satisfy Assumption 6 with exp-concavity factor of $C_{down}/C_{exp}$. So we can use the ProDR algorithm (Algorithm 1) to control the dynamic regret wrt losses $\ell_t$. Let $\beta = \min\{C_{down}/(2C_{exp}), 1/(8GB\sqrt{d})\}$ and $\gamma = \frac{1}{4\left(2GB\sqrt{d\beta} + 1/(2\sqrt{\beta})\right)^2}$ and $G' = 1 + GB\sqrt{d}C_{down}/C_{exp}$. Hence continuing from Eq.(7), we apply Theorem 3.1 to obtain

$$\mathbf{Reg}_T \leq \tilde{O}\left(d^3(T^{1/3}C_T^{2/3} \vee 1)\right).$$

This completes the proof of the theorem. ∎

*Remark* 3.9. Although noise distributions are known as we assumed, the coefficient of our regret upper bound depends highly on the distribution. As is indicated by Xu & Wang (2021), when the noise $N_t$ is an i.i.d. Gaussian noise with zero mean and $\sigma$ standard deviation, this coefficient is exponentially large w.r.t. $\frac{1}{\sigma}$ as $\sigma$ approaches 0, which is counter-intuitive.

### 3.3 Lower Bound on Dynamic Pricing Regret

So far, we have developed a ProDR algorithm that is suitable for domain-constraint optimization of generalized linear model, and have constructed a PDRP algorithm to solve the linear contextual pricing problem where PDRP achieves a $\tilde{O}(d^3(T^{\frac{1}{3}}C_T^{\frac{2}{3}} \vee 1))$ dynamic regret. This upper regret bound is optimal for online exp-concave optimization as is shown by Baby & Wang (2021), but is it still optimal for our feature-based dynamic pricing setting in specific? The answer is Yes. This dynamic regret is near-optimal up to $d$ and $\log T$ factors, and here we present the following theorem.

**Theorem 3.10** (Lower dynamic regret bound). *Let $d = 1$ in the contextual pricing problem we consider. For any algorithm $\mathcal{A}$, there exists a specific problem setting where $\mathcal{A}$ has to suffer an $\Omega(T^{\frac{1}{3}}C_T^{\frac{2}{3}} \vee 1)$ expected dynamic regret.*

With this theorem, we may claim that our PDRP algorithm is near-optimal. We here show a proof sketch and defer the full proof to Appendix B.6.

*Proof Sketch.* The proof is developed in three steps: Firstly, we construct a hypothesis set $\boldsymbol{\Theta}$ in which there are $N$ different $\{\theta_t^*\}_{t=1}^T$ series whose total variations are upper bounded by $C_T$. For any pair of two different series $\{\theta_t^*\}_{t=1}^T$'s in $\boldsymbol{\Theta}$, they are identical for $T/3$ out of $T$ rounds in total, and are different by some small $\delta$ for the rest $2T/3$ rounds. Secondly, we show that their corresponding feedback distributions are also "similar" to each other under the metric of KL-divergence. Therefore, according to Fano's Inequality, any algorithm can hardly distinguish among these distributions. Finally, we show that a failure of correctly distinguish the underlying distribution (i.e., the real $\{\theta_t^*\}_{t=1}^T$ series) will result in an $\Omega(T^{\frac{1}{3}}C_T^{\frac{2}{3}} \vee 1)$ regret. ∎

## 4 Conclusion

In this work, we studied the non-stationary contextual pricing problem under safety constraints. We first presented the ProDR algorithm for minimizing universal dynamic regret in the framework of proper OCO with co-variates and exp-concave losses. This contribution could be of independent interest in the context of non-stationary online learning. As a concrete application, we constructed our pricing algorithm, PDRP, by making use of ProDR as the base learner. We showed that PDRP attains a $\tilde{O}(d^3(T^{\frac{1}{3}}C_T^{\frac{2}{3}} \vee 1))$ dynamic regret in our pricing problem setting. Finally, we proved that this rate is information-theoretically optimal (modulo dependencies on $d$ and $\log T$).

## Acknowledgements

The authors are very grateful to the associate editor András György and three anonymous reviewers for their constructive comments. This research is partially supported by NSF Award #2007117, the Adobe Data Science Research Award and a start-up grant from the UCSB Department of Computer Science.

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

## A    Preliminaries

For the sake of completeness, we recall the description of the Follow-the-Leading-History (FLH) algorithm (see Hazan & Seshadhri, 2007).

---

FLH: inputs - Learning rate $\zeta$, $\mathcal{G}$, $\phi > 0$ and $T$ ONS base learners $E^1, \ldots, E^T$ initialized with parameters $G = \mathcal{G}$, $D = 2\phi\sqrt{d}$, $\alpha = \zeta$ and decision set $\mathcal{D} = \{\theta \in \mathbb{R}^d : \|\theta\|_\infty \leq \phi\}$. The learner $E_t$ starts operating from time $t$.

1. For each $t$, $v_t = (v_t^{(1)}, \ldots, v_t^{(t)})$ is a probability vector in $\mathbb{R}^t$. Initialize $v_1^{(1)} = 1$.
2. In round $t$, set $\forall j \leq t$, $\theta_t^j \leftarrow E^j(t)$ (the prediction of the $j^{th}$ base learner at time $t$). Play $\theta_t^{\text{alg}} = \sum_{j=1}^t v_t^{(j)} \theta_t^{(j)}$.
3. After receiving loss $f_t$, set $\hat{v}_{t+1}^{(t+1)} = 0$ and perform update for $1 \leq i \leq t$:

$$\hat{v}_{t+1}^{(i)} = \frac{v_t^{(i)} e^{-\zeta f_t(\theta_t^{(i)})}}{\sum_{j=1}^t v_t^{(j)} e^{-\zeta f_t(\theta_t^{(j)})}}$$

4. Addition step - Set $v_{t+1}^{(t+1)}$ to $1/(t+1)$ and for $i \neq t+1$:

$$v_{t+1}^{(i)} = (1 - (t+1)^{-1})\hat{v}_{t+1}^{(i)}$$

---

Figure 1: FLH algorithm

Next, we describe the Online Newton Step (ONS) algorithm (see Hazan et al., 2007).

---

ONS: inputs - Exp-concavity factor $\alpha$ and $G, D > 0$. Decision set $\mathcal{D}$.

1. At round 1, predict 0.
2. Let $\beta = \frac{1}{2}\min\{\frac{1}{4GD}, \alpha\}$. At iteration $t > 1$ predict:

$$w_t \in \underset{\theta \in \mathcal{D}}{\operatorname{argmin}} \|w_{t-1} - \frac{1}{\beta}A_{t-1}^{-1}\nabla_{t-1} - \theta\|_{A_{t-1}},$$

where $\nabla_i = \nabla f_i(w_i)$, $A_t = \frac{I_d}{\beta^2 D^2} + \sum_{i=1}^t \nabla_i \nabla_i^\top$.

---

Figure 2: ONS algorithm

## B    Detailed Proof

### B.1    Proof of Lemma 2.2

*Proof.* To begin with, we know that

$$h_t(\theta) = -\mathbb{1}_t \cdot \frac{f(\omega)}{1 - F(\omega)} + (1 - \mathbb{1}_t) \cdot \frac{f(\omega)}{F(\omega)},$$

where $\omega = v_t - x_t^\top\theta$. Since $\exists \theta_t \in \mathcal{D}_t$ such that $v_t = J(x_t^\top\theta_t)$, given that $J'(u) \in (0,1)$ (see Xu & Wang, 2021, Eq.(19)), we know that $\omega \in [J(-B) - B, J(B) + B]$ is bounded in a closed interval. Since we assume that $f(\omega) > 0, \forall \omega \in \mathbb{R}$, we know that $f_{\min} = \inf_{\omega \in [J(-B)-B, J(B)+B]} f(\omega) > 0$ and $F(\omega) \in [F(J(-B) - B), F(J(B) + B)] \subset (0,1)$. Remember that we denote $B_f := \sup_{\omega \in \mathbb{R}} f(\omega) < +\infty$. As a result, we have

$$0 < f_{\min} \leq \frac{f(\omega)}{1 - F(\omega)} \leq \frac{B_f}{1 - F(J(B) + B)} < +\infty$$

$$0 < f_{\min} \leq \frac{f(\omega)}{F(\omega)} \leq \frac{B_f}{F(J(-B) - B)} < +\infty.$$

Since $h_t(\theta) = \frac{f(\omega)}{1-F(\omega)}$ for $\mathbb{1}_t = 1$ or $h(t) = \frac{f(\omega)}{F(\omega)}$ for $\mathbb{1}_t = 0$, we know that $|h_t(\theta)| \in [f_{\min}, \frac{B_f}{\min\{1-F(J(B)+B),F(J(-B)-B)\}}]$. Let $h_{\max} = \frac{B_f}{\min\{1-F(J(B)+B),F(J(-B)-B)\}}$ and $h_{\min} = f_{\min}$, and the lemma is therefore proved. ∎

## B.2 Proof of Lemma 3.3

*Proof.* We have that for any $\theta \in \mathcal{D}_\infty^B$,

$$|\hat{\ell}_t(\theta) - \hat{\ell}_t(\hat{\theta}_t)| = \left|1/\sqrt{\beta} + \sqrt{\beta} \cdot \nabla \ell_t(\hat{\theta}_t)^\top (\theta - \hat{\theta}_t)\right| \cdot \left|\sqrt{\beta} \nabla \ell_t(\hat{\theta}_t)^\top (\theta - \hat{\theta}_t)\right|$$
$$\leq \left(1 + 2GB\beta\sqrt{d}\right) |\nabla \ell_t(\hat{\theta}_t)^\top (\theta - \hat{\theta}_t)|,$$

where in the last line we apply triangle inequality and the facts that $|\nabla \ell_t(\hat{\theta}_t)^\top (\theta - \hat{\theta}_t)| \leq G\|\theta - \hat{\theta}_t\|_2$ with $\|\theta - \hat{\theta}_t\|_2 \leq 2B\sqrt{d}$.

Putting $G' = 1 + 2GB\beta\sqrt{d}$ completes the lemma.

∎

## B.3 Proof of Lemma 3.4

*Proof.* For the simplicity of notation, we denote $\nabla_t := \nabla \ell_t(\hat{\theta}_t)$, and we have: $S_t(\theta) = \min_{x \in \mathcal{D}_t} |\nabla_t^\top (x - \theta)|$. Since $S_t(\theta)$ is convex in $\mathbb{R}^d$, we have:

$$S_t(\theta_2) \geq S_t(\theta_1) + \langle \nabla S_t(\theta_1), \theta_2 - \theta_1 \rangle, \forall \theta_1, \theta_2 \in \mathcal{D}_\infty^B.$$

Now we conduct an orthogonal decomposition: $\nabla S_t(\theta_1) = \mu_1 \nabla_t + \nabla_t^\perp$, where $\nabla_t^\top \nabla_t^\perp = 0$. Let $\theta_3 = \theta_2 + \mu_2 \nabla_t^\perp$, and we have $|\nabla_t^\top (x - \theta_2)| = |\nabla_t^\top (x - \theta_3)|, \forall x \in \mathbb{R}^d$. In other words, we have $S_t(\theta_2) = S_t(\theta_3)$ and therefore we have:

$$S_t(\theta_2) = S_t(\theta_3) \geq S_t(\theta_1) + \langle \nabla S_t(\theta_1), \theta_3 - \theta_1 \rangle$$
$$= S_t(\theta_1) + \langle \mu_1 \nabla_t + \nabla_t^\perp, \theta_2 + \mu_2 \nabla_t^\perp - \theta_1 \rangle$$
$$= S_t(\theta_1) + \langle \nabla S_t(\theta_1), \theta_2 - \theta_1 \rangle + \mu_2 \langle \nabla_t^\perp, \nabla_t^\perp \rangle, \forall \theta_2 \in \mathbb{R}^d, \mu_2 \in \mathbb{R}$$

In other words, $\mu_2\|\nabla_t^\perp\|_2^2 \leq S_t(\theta_2) - S_t(\theta_1) - \langle \nabla S_t(\theta_1), \theta_2 - \theta_1 \rangle$. Denote $\eta_1 = \operatorname{argmin}_{x \in \mathcal{D}_t} |\nabla_t^\top (x - \theta_1)|$, and $\eta_2 = \operatorname{argmin}_{x \in \mathcal{D}_t} |\nabla_t^\top (x - \theta_2)|$. Notice that

$$S_t(\theta_2) - S_t(\theta_1) = |\nabla_t^\top (\eta_2 - \theta_2)| - |\nabla_t^\top (\eta_1 - \theta_1)|$$
$$\leq |\nabla_t^\top (\eta_1 - \theta_2)| - |\nabla_t^\top (\eta_1 - \theta_1)|$$
$$\leq |\nabla_t^\top (\theta_1 - \theta_2)| \tag{8}$$
$$\leq \|\nabla_t\|_2 \cdot \|\theta_1 - \theta_2\|_2$$
$$\leq G \cdot \|\theta_1 - \theta_2\|_2.$$

Here the first inequality comes from the definition of $\eta_2$, the second inequality is an application of the triangular inequality, the third inequality is derived from Cauchy-Schwarz Inequality, and the last inequality is from Assumption A5 on the Lipschitzness of $\ell_t(\theta)$ over $\mathcal{D}_t$. Therefore, $S_t(\theta)$ is $G$-Lipschitz as well over $\mathcal{D}_\infty^B$, and we have:

$$\mu_2\|\nabla_t^\perp\|_2^2 \leq S_t(\theta_2) - S_t(\theta_1) - \langle \nabla S_t(\theta_1), \theta_2 - \theta_1 \rangle$$
$$\leq 2G\|\theta_2 - \theta_1\|_2.$$

This holds for any $\theta_1, \theta_2 \in \mathcal{D}_\infty^B$. However, we may fix $\theta_1$ and $\theta_2$ while also let $\mu_2 \to +\infty$ since it holds for any $\mu_2 \in \mathbb{R}$. If $\|\nabla_t^\perp\|_2 \neq 0$ then it will fall into a contradiction. Therefore, we know that $\nabla_t^\perp = \mathbf{0}$ and $\nabla S_t(\theta)$ is always in the same direction of $\nabla_t$.

Without losing generality, denote $\nabla S_t(\theta_1) := \lambda \cdot \nabla_t$. In the following, we will prove that $\lambda = \pm 1$ or $0$. From Eq. (8) line 3, we know that $S_t(\theta_2) - S_t(\theta_1) \leq |\nabla_t^\top(\theta_1 - \theta_2)|$. Combined with the convexity of $S_t(\theta)$, we have:

$$
\begin{aligned}
|\nabla_t^\top(\theta_1 - \theta_2)| \geq &S_t(\theta_2) - S_t(\theta_1) \\
\geq &\nabla S_t(\theta_1)^\top(\theta_2 - \theta_1) \\
= &\lambda \cdot \nabla_t^\top(\theta_2 - \theta_1).
\end{aligned}
\tag{9}
$$

Notice that we can choose arbitrary $\theta_2$ without changing $\lambda$, we may let $\theta_2 = 0$ and $\theta_2 = 2\theta_1$ in Eq. (9):

$$
\pm\lambda \cdot \nabla_t^\top\theta_1 \leq |\nabla_t^\top\theta_1|
$$

If $\nabla_t^\top\theta_1 \neq 0$, then we have $\lambda \in [-1, 1]$. Otherwise, we know from Eq. (9) that $|\nabla_t^\top\theta_2| \geq \lambda \cdot \nabla_t^\top\theta_2, \forall\theta_2$, and similarly we have $\lambda \in [-1, 1]$. Now we denote $\theta_4 := \frac{\theta_1 + \eta_1}{2}$, and we have:

$$
\langle\nabla S_t(\theta_1), \theta_4 - \theta_1\rangle + S_t(\theta_1) \leq S_t(\theta_4)
\tag{10}
$$

from the convexity of $S_t$. And we also have:

$$
\begin{aligned}
S_t(\theta_4) &= \min_{x \in \mathcal{D}_t} |\nabla_t^\top(x - \theta_4)| \\
&\leq |\nabla_t^\top(\eta_1 - \theta_4)| \\
&= |\nabla_t^\top\frac{\theta_1 - \eta_1}{2}| \\
&= \frac{1}{2}S_t(\theta_1) \\
&= |\nabla_t^\top(\theta_1 - \theta_4)| \\
&= S_t(\theta_1) - |\nabla^\top(\theta_1 - \theta_4)|.
\end{aligned}
\tag{11}
$$

Combine Eq. (10) and (11), we have:

$$
\langle\nabla S_t(\theta_1), \theta_4 - \theta_1\rangle \leq S_t(\theta_4) - S_t(\theta_1) = -|\nabla_t^\top(\theta_1 - \theta_4)|
\tag{12}
$$

Plug in $\nabla S_t(\theta_1) = \lambda\nabla_t$ to Eq. (12), and we have:

$$
\lambda \cdot \nabla_t^\top(\theta_4 - \theta_1) \leq -|\nabla_t^\top(\theta_1 - \theta_4)|.
\tag{13}
$$

According to Eq. (13), if $\nabla_t^\top(\theta_4 - \theta_1) > 0$, then we have $\lambda \leq -1$; if $\nabla_t^\top(\theta_4 - \theta_1) < 0$, then we have $\lambda \geq 1$. Since we already know that $\lambda \in [-1, 1]$, then for the two case we should have $\lambda = -1$ or $\lambda = 1$.

Finally, what if $\nabla_t^\top(\theta_4 - \theta_1) = 0$? In this case, it means that $\nabla_t^\top(\eta_1 - \theta_1)/2 = 0$. Since $\eta_1 = \mathrm{argmin}_{x \in \mathcal{D}_t} |\nabla_t^\top(x - \theta_1)|$, we know that $S_t(\theta_1) = 0$ at this time. Since $S_t(\theta) \geq 0, \forall\theta \in \mathbb{R}^d$, we know that $S_t(\theta) \geq S_t(\theta_1) + \mathbf{0}^\top(\theta - \theta_1)$ and as a result $0 \in \partial S_t(\theta_1)$. This in fact holds the lemma.

■

### B.4 Proof of Lemma 3.5

*Proof.* We begin by noticing that $\hat{\ell}_t(\theta)$ is exp-concave over $\mathcal{D}_\infty^B$. To see this, note that by the triangular inequality and Cauchy-Schwarz Inequality,

$$
|\nabla\ell_t(\hat{\theta}_t)^\top(\theta - \hat{\theta}_t)\sqrt{\beta} + 1/(2\sqrt{\beta})| \leq |\nabla\ell_t(\hat{\theta}_t)^\top(\theta - \hat{\theta}_t)|\sqrt{\beta} + 1/(2\sqrt{\beta}) \leq 2GB\sqrt{d\beta} + 1/(2\sqrt{\beta}),
$$

where we use the fact that $\|\nabla\ell_t(\hat{\theta}_t)\|_2 \leq G$ by Assumption A5 and $\|\theta - \hat{\theta}_t\|_2 \leq 2B\sqrt{d}$ as $\theta \in \mathcal{D}_\infty^B$ and $\hat{\theta}_t \in \mathcal{D}_t \subset \mathcal{D}_\infty^B$.

With $\gamma$ as defined in the statement of the lemma, we have that the losses $\hat{\ell}_t(\theta)$ are $2\gamma$ exp-concave over $\mathcal{D}_\infty^B$. (see Cesa-Bianchi & Lugosi, 2006, Section 3.3).

Now we proceed to show that the losses $f_t(\theta)$ are in-fact exp-concave with appropriate exp-concavity factor.

For the sake of brevity, let us denote

$$\nabla \hat{\ell}_t(u) = 2\sqrt{\beta}\left(\nabla\ell_t(\hat{\theta}_t)^\top(u - \hat{\theta}_t)\sqrt{\beta} + \frac{1}{2\sqrt{\beta}}\right)\nabla\ell_t(\hat{\theta}_t)$$
$$:= p(u)\nabla\ell_t(\hat{\theta}_t).$$

We have that for any $u, v \in \mathcal{D}_\infty^B$,

$$\hat{\ell}_t(v) \geq \hat{\ell}_t(u) + p(u)\nabla\ell_t(\hat{\theta}_t)^\top(v - u)$$
$$+ \gamma\left(p(u)\nabla\ell_t(\hat{\theta}_t)^\top(v - u)\right)^2. \tag{14}$$

Due to convexity, we have

$$S_t(v) \geq S_t(u) + \lambda\nabla\ell_t(\hat{\theta}_t)^\top(v - u), \tag{15}$$

for some $\lambda \in \{-1, 0, 1\}$ as per Lemma 3.4.

Adding Eq.(14) and (15), we obtain

$$f_t(v) \geq f_t(u) + \nabla f_t(u)^\top(u - v)$$
$$+ \gamma p(u)^2\left(\nabla\ell_t(\hat{\theta}_t)^\top(v - u)\right)^2$$
$$= f_t(u) + \nabla f_t(u)^\top(u - v)$$
$$+ \gamma\left(\frac{p(u)}{\lambda + p(u)}\right)^2\left(\nabla f_t(u)^\top(v - u)\right)^2. \tag{16}$$

Next, we proceed to obtain a lower bound on the exp-concavity factor. Note that

$$p(u) \geq 2\sqrt{\beta}\left(-2GB\sqrt{d\beta} + \frac{1}{2\sqrt{\beta}}\right) \geq 2\sqrt{\beta} \cdot \frac{1}{4\sqrt{\beta}} = \frac{1}{2}$$

where the first inequality is via Cauchy-Schwarz Inequality and the second inequality holds due to the fact that $\beta \leq 1/(8GB\sqrt{d})$ due to the setting in Theorem 3.1

Similarly we have that

$$|p(u) + \lambda| \leq 4GB\beta\sqrt{d} + 2 \leq 5/2,$$

where in the last line we used $\beta \leq 1/(8GB\sqrt{d})$.

Combining the last two displays, we have that

$$\gamma\left(\frac{p(u)}{\lambda + p(u)}\right)^2 \geq \gamma/25.$$

Applying this lower bound to Eq.(16) now yields the exp-concavity of $f_t(\theta)$ claimed in the lemma.

Next, we proceed to calculate the Lipschitz constant of $f_t$. Since $\|\nabla\ell_t(\hat{\theta}_t)\|_2 \leq G$, by Lemma 3.4 we conclude that $G'S_t(\theta)$ is $G'G$ Lipschitz in L2 norm across $\mathbb{R}^d$. Now using Lemma 3.3 we conclude that the losses $f_t$ are $2G'G$ are Lipschitz in L2 norm across $\mathcal{D}_\infty^B$.

■

## B.5  Proof of Lemma 3.8

Xu & Wang (2021, Lemma 7) has proved the $\frac{C_{down}}{C_{exp}}$-exp-concavity. Here we prove the other claim on Lipschitzness.

*Proof.* Notice that $\ell_t(\theta)$ is a continuous function. Therefore, for any $\theta_1, \theta_2 \in \mathcal{D}_t$, there exists a $\theta_3 = \epsilon\theta_1 + (1-\epsilon)\theta_2$ for some $\epsilon \in [0,1]$ such that

$$
\begin{aligned}
\ell_t(\theta_1) - \ell_t(\theta_2) &= \nabla\ell_t(\theta_3)^\top(\theta_1 - \theta_2) \\
&= h_t(\theta_3)x_t^\top(\theta_1 - \theta_2) \\
&\leq h_{\max}\|x_t\|_2\|\theta_1 - \theta_2\|_2 \\
&= h_{\max}\|\theta_1 - \theta_2\|_2 \\
&= G\|\theta_1 - \theta_2\|_2
\end{aligned}
\tag{17}
$$

where $h_{\max}$ is defined in Appendix B.1. In Eq.(17), the first equality is by Lagrange interpolation, the second equality is by definition of $h_t(\theta)$, the third inequality is by Cauchy-Schwarz Inequality, the fourth equality is by the assumption that $x_t \in \mathcal{D}_2^1$, and the last inequality is from the fact that $h_{\max} = G$. Since $\mathcal{D}_t$ is convex, we know that $\theta_3 \in \mathcal{D}_t$. Therefore, the lemma is proven.

∎

## B.6  Lower Bound Proof (Proof of Theorem 3.10)

Here we present and prove the following theorem, which is sufficient to show a $\Omega(T^{\frac{1}{3}}C_T^{\frac{2}{3}})$ lower bound for $C_T > \frac{1}{\sqrt{T}}$.

**Theorem B.1.** *Consider a feature-based dynamic pricing problem with $d = 1, x_t = 1, N_t \sim_{i.i.d.} \mathcal{N}(0,1), t = 1, 2, \ldots, T$ and $C_T > \frac{1}{\sqrt{T}}$ For any algorithm $\mathcal{A}$ there exists a specific setting such that $\mathcal{A}$ suffer $\Omega(T^{\frac{1}{3}}C_T^{\frac{2}{3}})$ expected regret even with $y_t$ observable.*

The sufficiency comes from the fact that we cannot observe any realized $y_t$'s in the pricing problem (but a binary indicator instead). In comparison, the lower bound in Theorem B.1 even works for observable $y_t$'s, which is sufficient to derive the binary feedback (by comparing $y_t$ with $v_t$).

*Proof.* To summarize the procedure of proof: Denote $[n] := \{1, 2, \ldots, n\}$ for any positive integer $n$. Define $\theta_0 = 1, \theta_1 = 1 + \delta(T, C_T)$ where $\delta = \frac{1}{40}(\frac{C_T}{T})^{\frac{1}{3}}$ is an additional amount. Then we construct a set $S \subset \{0,1\}^T$ consisting of randomly-sampled $\beta^{(i)} \in \{0,1\}^T, i = 1, 2, \ldots, N$ that we will use to construct $\theta_t^*(i)$ series (each $i$ indicating a specific $\{\theta_t^*\}$ series) later. Afterward, we will show that the $\{\theta_t^*(i)\}$ and the $\{\theta_t^*(j)\}$ series are hard to distinguish by any algorithm, and we will further show that a large enough regret caused by this misspecification. In this way, we can prove an expected lower regret bound (where the expectation is also taken over different $\{\theta_t^*(i)\}$).

The process to sample each $\beta^{(i)}$ is as follows: We split $[T]$ uniformly into $m = \frac{C_T}{4\delta}$ intervals, with each length $\frac{4T\delta}{C_T}$. Since $\delta = \frac{1}{40}(\frac{C_T}{T})^{\frac{1}{3}}$ and $C_T \geq \frac{1}{\sqrt{T}}$, we know that $m \geq 10$. Denote these intervals as $I_1, I_2, \ldots, I_m$. For any $\beta^{(i)} \in S$, we construct it in a stochastic process: For each index interval $I_k, k = 1, 2, \ldots, m$, we generate a random variable $Z_k^{(i)} \sim Ber(\frac{1}{2})$ independently, and then let $\beta_l^{(i)} = Z_k^{(i)}, \forall l \in I_k$. Denote the vector $Z^{(i)} = [Z_1^{(i)}, Z_2^{(i)}, \ldots, Z_m^{(i)}]^\top \in \{0,1\}^m$, and we know that $\mathbb{E}[\|Z^{(i)} - Z^{(j)}\|_1] = \frac{m}{2}$. Accordingly, we have $\mathbb{E}[\|\beta^{(i)} - \beta^{(j)}\|_1] = \frac{m}{2} \cdot \frac{4T\delta}{C_T} = \frac{T}{2}$.

Therefore, according to Hoeffding's inequality, we have:

$$
\begin{aligned}
&\Pr[|\|Z^{(i)} - Z^{(j)}\|_1 - \frac{m}{2}| \leq \frac{m}{6}] \geq 1 - 2 \cdot e^{-\frac{(\frac{m}{6})^2}{2m}} \\
&\Leftrightarrow \Pr[|\|\beta^{(i)} - \beta^{(j)}\|_1 - \frac{T}{2}| \leq \frac{T}{6}] \geq 1 - 2 \cdot e^{-\frac{m}{72}}, \forall i, j \in [N].
\end{aligned}
\tag{18}
$$

By applying a union bound over all $\binom{N}{2}$ pairs of $i, j \in [N]$, we have:

$$\Pr[|\|\beta^{(i)} - \beta^{(j)}\|_1 - \frac{T}{2}| \leq \frac{T}{6}, \forall i, j \in [N]] \geq 1 - N^2 \cdot e^{-\frac{m}{72}}.$$

Also, we know that $\Pr[\beta^{(i)} \neq \beta^{(j)}] = \Pr[Z^{(i)} \neq Z^{(j)}] = 1 - \frac{1}{2^m}$ for $i \neq j$. By applying a union bound over all $\binom{N}{2}$ pairs of $i, j$, we have $\Pr[\beta^{(i)} \neq \beta^{(j)}] \geq 1 - \frac{N^2}{2^{m+1}}$. Combining these two probability bounds, we know that in this way we can find a satisfactory set $S$ with probability at least $\Pr \geq 1 - N^2 \cdot (e^{-\frac{m}{72}} + 2^{-(m+1)})$. Let $N = e^{\frac{m}{200}}$ (and therefore $\log N = \frac{m}{200} = \frac{C_T}{800\delta}$), and then $\Pr \geq 1 - N^2 \cdot (e^{-\frac{m}{72}} + 2^{-(m+1)}) \geq 1 - (e^{-\frac{m}{300}} + e^{-\frac{3}{5}m})$. Since the total number of possible $S$ (i.e., any set consisting $N$ (repeatable) vectors $\beta \in \{0,1\}^T$) is $(2^m)^N$ and we are uniformly sampling from this whole family, the expected total number of satisfactory $S$ is at least $(2^m)^N \cdot (1 - (e^{-\frac{m}{300}} + e^{-\frac{3}{5}m}))$. Since $m \geq 10$ as we showed above, we have $(2^m)^N \cdot (1 - (e^{-\frac{m}{300}} + e^{-\frac{3}{5}m})) \geq 2^{10 \times 1} \cdot (1 - e^{-\frac{1}{30}} - e^{-6}) = 31.0325 > 1$. As a result, there must exist at least one satisfactory $S$ in the whole possible set family, such that: (1) $\frac{T}{3} \leq \|\beta^{(i)} - \beta^{(j)}\|_1 \leq \frac{2T}{3}$, and (2) $\beta^{(i)} \neq \beta^{(j)}, \forall i \neq j \in [N]$. We here pick this satisfactory $S$ and in the following we use it for further proof.

Now, for each $\beta^{(i)} \in S$, we generate a sequence of parameter $\{\theta_t^*(i)\}$ according to $\beta^{(i)}$: For $t = 1, 2, \ldots, T$, we let $\theta_t^*(i) = 1 + \delta \cdot \beta_t^{(i)}$, i.e., if $\beta^{(i)} = 0$, then $\theta_t^*(i) = \theta_0 = 1$; if $\beta^{(i)} = 1$, then $\theta_t^*(i) = 1 + \delta$. Therefore, we have the following result:

$$\mathrm{TV}(\{\theta_t^*(i)\}) \leq m \cdot \delta = \frac{C_T}{4} < C_T.$$

This is because $\|\theta_t^*(i) - \theta_{t+1}^*(i)\| > 0$ only if $\exists k \in [m]$ s.t. $t \in I_k, t+1 \in I_{k+1}$. As a result, the total variation of this $\{\theta_t^*(i)\}$ satisfies the upper bound $C_T$.

Now, let us consider the realized valuation sequence $\{y_t\}$. For any $i \in [N]$, denote

$$\mathbf{y}(i) := [x_1(1 + \beta_1^{(i)}\delta) + N_1, x_2(1 + \beta_2^{(i)}\delta) + N_2, \ldots, x_T(1 + \beta_T^{(i)}) + N_T]^\top$$

Let us denote the distribution of $\mathbf{y}(i)$ as $\mathbb{P}_i, i = 1, 2, \ldots, N$. Recall that $x_t = 1$ and $N_t \sim \mathcal{N}(0,1), \forall t$, and we have $\mathbb{P}_i = [\mathcal{N}(1 + \beta_1^{(i)}\delta, 1), \mathcal{N}(1 + \beta_2^{(i)}\delta, 1), \ldots, \mathcal{N}(1 + \beta_T^{(i)}\delta)]^\top$. Consider the difference between $\mathbb{P}_i$ and $\mathbb{P}_j$ while fixing $\beta^{(i)}$ and $\beta^{(i)}$, and for any $i, j \in [N], i \neq j$ we have:

$$
\begin{aligned}
KL[\mathbb{P}_i || \mathbb{P}_j] &= \sum_{t=1}^{T} KL[\mathcal{N}(1 + \beta_t^{(i)}\delta, 1) || \mathcal{N}(1 + \beta^{(j)}\delta, 1)] \\
&= \sum_{t=1}^{T} \frac{(\beta_t^{(i)} - \beta_t^{(j)})^2 \delta^2}{2} \\
&= \frac{\delta^2}{2} \cdot \|\beta^{(i)} - \beta^{(j)}\|_2^2 \\
&= \frac{\delta^2}{2} \cdot \|\beta^{(i)} - \beta^{(j)}\|_1.
\end{aligned}
\tag{19}
$$

Again, the KL-divergence is conditioning on $\beta^{(i)}$ and $\beta^{(j)}$. Here the first line is from the fact that $y_t$'s are independent for every $t$, the second line is by $x_t = 1$, the third line is from the fact that $KL[\mathcal{N}(\mu_1, \sigma_1) || \mathcal{N}(\mu_2, \sigma_2)] = \log \frac{\sigma_2}{\sigma_1} + \frac{\sigma_1^2 + (\mu_1 - \mu_2)^2}{2\sigma_2^2} - \frac{1}{2}$, the fourth line is by calculation and the fifth line is from that $|\beta_t^{(i)} - \beta_t^{(j)}| \in \{0, 1\}$.

Here we introduce a Fano's Inequality as the following proposition:

**Proposition B.2** (Fano's Inequality). *Let $X_1, X_2, \ldots, X_n \sim_{i.i.d.} \mathbb{P}$ where $\mathbb{P} \in \{\mathbb{P}_1, \mathbb{P}_2, \ldots, \mathbb{P}_N\}$ is a distribution family. Let $\Psi$ be any function of $X_1, X_2, \ldots, X_n$ taking values in $\{1, 2, \ldots, N\}$. Let $\alpha = \max_{j \neq k} KL(\mathbb{P}_j || \mathbb{P}_k)$.[2] Then*

$$\frac{1}{N} \sum_{j=1}^{N} \mathbb{P}_j(\Psi \neq j) \geq 1 - \frac{n\alpha + \log 2}{\log N}.$$

---

[2]Usually it is denoted as $\beta$, but here we denote it as $\alpha$ for clarity, since we have already defined $\beta^{(i)}$ as vectors in $S$.

According to Fano's Inequality, we have:

$$\inf_{\Psi:\mathbb{R}^T \to \{1,2,\dots,N\}} \sup_{i\in\{1,2,\dots,N\}} \mathbb{P}_i(\Psi \neq i) \geq \inf_{\Psi} \frac{1}{N}\sum_{i=1}^{N} \mathbb{P}_i(\Psi \neq i) \geq 1 - \frac{n\alpha + \log 2}{\log N} \geq \frac{1}{2} - \frac{n\alpha}{\log N}. \tag{20}$$

Here $n = 1$ since only one specific $\mathbf{y}(i)$ covers the whole time series and is only sampled once, and $\alpha = \max_{i,j\in[N],i\neq j} KL[\mathbf{y}(i)\|\mathbf{y}(j)] = \max_{i,j\in[N],i\neq j} \frac{\delta^2}{2} \cdot \|\beta^{(i)} - \beta^{(j)}\|_1 \leq \frac{\delta^2 T}{3}$ is the upper bound of KL-divergences on different distributions. Now we specify the function $\Psi_{\mathcal{A}}$ for any pricing algorithm $\mathcal{A}$: At each round $t = 1, 2, \dots, T$, suppose the algorithm $\mathcal{A}$ proposes a price $v_t^{\mathcal{A}}$. Define a vector $\mathbf{w} = [w_1, w_2, \dots, w_T]^\top$ where $w_t = \mathbb{1}[v_t^{\mathcal{A}} \geq \frac{J(\theta_0)+J(\theta_1)}{2}]$ is a Boolean value. Then we let $\Psi_{\mathcal{A}} = \operatorname{argmin}_i \in [N]\|\mathbf{w} - \beta^{(i)}\|_1$. Therefore, we have:

$$\begin{aligned}
2 \cdot \|\mathbf{w} - \beta^{(j)}\|_1 &\geq \|\beta^{(\Psi_{\mathcal{A}})} - \mathbf{w}\|_1 + \|\mathbf{w} - \beta^{(j)}\|_1 \\
&\geq \|\beta^{(\Psi_{\mathcal{A}})} - \beta^{(j)}\|_1, \forall j \in [N], j \neq \Psi_{\mathcal{A}} \\
&\geq \frac{T}{6}
\end{aligned}$$

Here the first inequality is from the optimality of $\Psi_{\mathcal{A}}$, the second inequality is from the triangular inequality, and the third inequality is from the Hoeffding bound in Eq. (18). Therefore we know that if $\Psi_{\mathcal{A}} \neq i$ then we have $\|\mathbf{w} - \beta^{(i)}\|_1 \geq \frac{T}{12}$, which further leads to

$$\begin{aligned}
&\sum_{t=1}^{T}(v_t^{\mathcal{A}} - J(x_t\theta_t^*(i)))^2 \\
&\geq \sum_{t=1}^{T}(\mathbb{1}[w_t = 1]\mathbb{1}[\beta_t^{(i)} = 0] + \mathbb{1}[w_t = 0]\mathbb{1}[\beta_t^{(i)} = 1])(v_t^{\mathcal{A}} - J(x_t\theta_t^*(i)))^2 \\
&= \sum_{t=1}^{T}\mathbb{1}[v_t^{\mathcal{A}} \geq \frac{J(\theta_0)+J(\theta_1)}{2}]\mathbb{1}[\beta_t^{(i)} = 0](v_t^{\mathcal{A}} - J(\theta_0))^2 + \mathbb{1}[v_t^{\mathcal{A}} < \frac{J(\theta_0)+J(\theta_1)}{2}]\mathbb{1}[\beta_t^{(i)} = 1](J(\theta_1 - v_t^{\mathcal{A}}))^2 \\
&\geq \sum_{t=1}^{T}\mathbb{1}[|w_t - \beta_t^{(i)}| = 1](\frac{J(\theta_1) - J(\theta_0)}{2})^2 \\
&= \|\mathbf{w} - \beta^{(i)}\|_1(\frac{J(\theta_1) - J(\theta_0)}{2})^2 \\
&\geq \frac{T}{12} \cdot (\frac{J(\theta_0) - J(\theta_1)}{2})^2.
\end{aligned}$$

Here the first line is because we omit the case where $\mathbb{1}[w_t = \beta_t^{(i)}]$, the second line is from the definition of $w_t$, the third line is from the facts of $\theta_0 < \theta_1$ and $J'(u) > 0, \forall u \in \mathbb{R}$, the fourth line is by the definition of $L_1$-norm and the last line is from the fact we mentioned prior to this equation. Now we propose a lemma of properties:

**Lemma B.3** (Properties of $g(v, u)$ and $J(u)$). *For $g(v, u)$ and $J(u)$ with $N_t \sim \mathcal{N}(0, 1)$, we have:*

1. *$J(u) > u$ when $u \in (0, \sqrt{\frac{\pi}{2}})$ and $J(u) < u$ when $u \in (\sqrt{\frac{\Pi}{2}}, +\infty)$.*

2. *$\exists c_J > 0$ s.t. $J'(u) \geq c_J, \forall u \in [-B, B]$.*

3. *$\exists c_g > 0$ s.t. $g(J(u), u) - g(v, u) \geq c_g(J(u) - v)^2, \forall v \in [0, B + J(B)]$.*

We will show the proof of Lemma B.3 by the end of this section. With Lemma B.3, when $\Psi_A \neq i$, we have:

$$
\begin{aligned}
\mathbf{Reg}_{\mathcal{A}} &= \sum_{t=1}^{T} g(J(x_t\theta_t^*(i)), x_t\theta_t^*(i)) - g(v_t^{\mathcal{A}}, x_t\theta_t^*(i)) \\
&\geq \sum_{t=1}^{T} c_g(v_t^{\mathcal{A}} - J(x_t\theta_t^*(i)))^2 \\
&\geq c_g \cdot \frac{T}{12} \cdot \left(\frac{J(\theta_0) - J(\theta_1)}{2}\right)^2 \\
&\geq c_g \cdot \frac{T}{12} \cdot \frac{c_J^2}{4} \cdot (\theta_1 - \theta_0)^2 \\
&\geq \frac{c_g c_J^2 \cdot T\delta^2}{48}.
\end{aligned}
\tag{21}
$$

Finally, let $\delta = \frac{1}{40}(\frac{C_T}{T})^{\frac{1}{3}}$, and according to Eq. (19),(20) and (21), we have:

$$
\begin{aligned}
\mathbb{E}[\mathbf{Reg}_{\mathcal{A}}] &\geq \sup_{i \in [N]} \mathbb{P}_i(\Psi_{\mathcal{A}} \neq i) \cdot \left(\sum_{t=1}^{T} g(J(x_t\theta_t^*(i)), x_t\theta_t^*(i)) - g(v_t^{\mathcal{A}}, x_t\theta_t^*(i))\right) \\
&\geq \sup_{i \in [N]} \mathbb{P}_i(\Psi_{\mathcal{A}} \neq i) \cdot \frac{c_g c_J^2 \cdot T\delta^2}{48} \\
&\geq \left(\frac{1}{2} - \frac{n\alpha}{\log N}\right) \cdot \frac{c_g c_J^2 \cdot T\delta^2}{48} \\
&= \left(\frac{1}{2} - \frac{\frac{\delta^2 T}{3}}{\frac{C_T}{800\delta}}\right) \cdot c_g c_J^2 \frac{\cdot T\delta^2}{48} \\
&= c_g c_J^2 \left(\frac{1}{2} - \frac{800}{3} \cdot \frac{\delta^3 T}{C_T}\right) \cdot \frac{\cdot T\delta^2}{48} \\
&= \frac{c_g c_J^2}{48}\left(\frac{1}{2} - \frac{1}{240}\right) \frac{T \cdot (\frac{C_T}{T})^{\frac{2}{3}}}{144} \\
&\geq \frac{c_g c_J^2}{307200} \cdot C_T^{\frac{2}{3}} T^{\frac{1}{3}}.
\end{aligned}
$$

This holds the theorem. ∎

Also, since our upper regret bound w.r.t. $T$ and $C_T$ is $\tilde{O}(1)$ when $C_T \leq \frac{1}{\sqrt{T}}$, which is trivial up to $\log T$ and $d$ factors, we may conclude that our upper regret bound of $\tilde{O}(T^{\frac{1}{3}}C_T^{\frac{2}{3}} \vee 1)$ is optimal with respect to $T$ and $C_T$.

*Proof of Lemma B.3.* We here prove each of them.

1. According to Xu & Wang (2021, Lemma 14), we know that $u - J(u)$ is monotonically increasing since $J'(u) \in (0, 1)$. Also, since $\frac{\partial g(v,u)}{\partial v}|_{v=J(u)} = 1 - F(J(u) - u) - J(u) \cdot f(J(u) - u) = 0$, we have $J(\sqrt{\frac{\pi}{2}}) = \sqrt{\frac{\pi}{2}}$. Therefore, $u - J(u) > 0$ when $u > \sqrt{\frac{\pi}{2}}$ and $u - J(u) < 0$ when $0 < u < \sqrt{\frac{\pi}{2}}$.

2. From Xu & Wang (2021, Appendix B.2.1.), we know that $J'(u) = 1 + \frac{1}{\phi'(\phi^{-1}(u))} \in (0, 1), \forall u \in \mathbb{R}$ where $\phi(\omega) = \frac{1 - F(\omega)}{f(\omega)} - \omega$ is invertible and smooth for standard Gaussian distribution. Therefore, we know that $J'(u)$ is continuous. Therefore, $\exists c_J > 0$ such that $\inf_{u \in [-B,B]} J'(u) = c_J$.

3. From the optimality of $J(u)$ we know that $\frac{\partial g(v,u)}{\partial v}|_{v=J(u)} = 1 - F(J(u) - u) - J(u) \cdot f(J(u) - u) = 0$. Define $q(u) := 1 - F(J(u) - u) - J(u) \cdot f(J(u) - u)$. Since $q(u) = 0, \forall u \in \mathbb{R}$, we have:

$$\frac{\partial q(u)}{\partial u} = 0$$
$$\Leftrightarrow \left( J'(u)(J(u)^2 - u \cdot J(u) - 2) - (J(u)^2 - u \cdot J(u) - 1) \right) f(J(u) - u) = 0$$
$$\Leftrightarrow J'(u) = 1 + \frac{1}{J(u)^2 - u \cdot J(u) - 2}.$$

The second line is by standard Gaussian noises and some calculations, and the third line is from the fact that $f(x) > 0$ for standard Gaussian distribution. Since we already know that $J'(u) \in (0, 1)$, we may then realized that $J(u)^2 - u \cdot J(u) - 2 < -1$. Notice that $\frac{\partial^2 g(v,u)}{\partial v^2} = (v^2 - vu - 2)f(v - u)$ for standard gaussian noise. Therefore, we have $\frac{\partial^2 g(v,u)}{\partial v^2} = (J(u)^2 - u \cdot J(u) - 2)f(J(u) - u) \le (-1) \cdot f_{\min} < 0$ where $f_{\min}$ has been defined in Appendix B.1 as the universal lower bound of $f$. This means that $g(v, u)$ is $f_{\min}$-strongly concave at $v = J(u)$, which further leads to the fact that there exists a neighborhood $v \in [J(u) - B_u, J(u) + B_u]$ with constant[3] $B_u$ such that $\frac{\partial^2 g(v,u)}{\partial v^2} \le -\frac{f_{\min}}{2}$. As a result, for $v \in [J(u) - B_u, J(u) + B_u]$ we have

$$g(J(u), u) - g(v, u) = -\frac{\partial g(v,u)}{\partial v}|_{v=J(u)}(J(u) - u) - \frac{1}{2} \cdot \frac{\partial^2 g(v,u)}{\partial v^2}|_{v=v' \in [J(u),v] \text{ or } [v,J(u)]}(J(u) - v)^2$$
$$\ge -\frac{1}{2}(-\frac{f_{\min}}{2})(J(u) - v)^2$$
$$= \frac{f_{\min}}{4}(J(u) - v)^2.$$

Now, let us consider the case when $v \in [0, B + J(B)]$ but $v \notin [J(u) - B_u, J(u) + B_u]$. On the one hand, $(J(u) - v)^2 \le (B + J(B) - (-B))^2 = (2B + J(B))^2$. On the other hand, $g(J(u), u) - g(v, u) \ge g(J(u), u) - \max\{g(J(u) - B_u, u), g(J(u) + B(u), u)\} > 0$. Denote $c_u := \inf_{u \in [-B,B]}\{g(J(u), u) - \max\{g(J(u) - B_u, u), g(J(u) + B(u), u)\}\}$, and we have $c_u > 0$. Therefore, we have:

$$g(J(u), u) - g(v, u) \ge c_u \ge \frac{c_u}{(2B + J(B))^2}(2B + J(B))^2 \ge \frac{c_u}{(2B + J(B))^2}(J(u) - v)^2.$$

Finally, let $c_g = \min\{\frac{f_{\min}}{4}, \frac{c_u}{(2B+J(B))^2}\}$, and we have proved the lemma.

∎

---

[3] $B_u$ can be defined as the inferior of all $B_u$ over all $u \in [-B, B]$ and is still a positive constant.

