# OpenReview forum: "Non-Stationary Contextual Pricing with Safety Constraints"
_TMLR — Accepted by TMLR_

### Review · Reviewer_1vaT · 2022-07-18

**Summary Of Contributions:**

This paper studies an interesting problem called non-stationary contextual pricing, a natural variant of the standard pricing problem that allows the best pricing strategy to change over time. The setting is of course interesting and important, as it can capture a broader range of pricing applications, particularly those requiring modeling the non-stationarity of the users' interests. This paper studies the problem from the lens of **universal dynamic regret**, an important concept in the online learning literature. To be specific, the authors first formulate this pricing problem (under certain acceptable assumptions) as an online learning problem with exp-concave loss. Although prior work has been studied in this regard, their algorithms are not **proper** in the sense that the decision may lie out of the range of the feasible set, which is not favored in this paper -- which is called the "safety constraint". To address so, they have to design a new algorithm with a provable guarantee that the decisions must lie in the feasible set, and also the algorithm can enjoy nice universal dynamic regret. The major technical ingredient is an interesting "transfer theorem" with the cooperation of the reduction scheme [Cutkosky & Orabona, 2018]. The important caveat in the trick is that the surrogate loss should be able to keep the exp-concavity such that the reduced problem can be handled by earlier studies developed for the exp-concave online learning with $L_{\infty}$ ball decision set.

To summarize, I believe the paper studies an interesting problem and contributes nice insight and technique to the community. The overall paper writing is of high quality (though some proofreadings are required), so I recommend the acceptance.




**Broader Impact Concerns:**

The paper mainly contributes to the theoretical aspect, so I cannot find foreseeable ethical issues.

**Requested Changes:**

Please check my comments in the last question (Strengths And Weaknesses).

**Strengths And Weaknesses:**

## Strengths
As mentioned in the last question (Summary Of Contributions), this paper studies an interesting and novel problem that is of practical appeal and requires certain theoretical innovations. The paper writing is generally good -- the logic flow is very easy to follow, and the contributions and difficulties are stated very clearly. I roughly check the proof and didn't find any flaw, and I believe the results should be correct after a high-level check of the proofs.

The explanation of using \ell_1-norm at the beginning of page 2 is highly appreciated.

## Weakness
My first concern is about the results in Section 3.4, which looks like a "bandit" variant of the other part (the standard one looks like a "full-information" setting). Most of the contents are placed in Appendix C, and actually, I am quite confused about the algorithm presented in this part due to the lack of introduction of ADA-ILTCB+ (Chen et al., 2019). For example, the authors should at least introduce what the algorithm is, what the inputs should be, and what the output and guarantee the algorithm will produce. Current presentations only confuse the reader and make the theoretical claims hard to convince the reader.

Another question in this Section 3.4: there is one black-box approach for non-stationary stochastic bandits learning/RL (https://arxiv.org/abs/2102.05406), which should be (conceptually) easier and more general than (Chen et al., 2019). The authors may have a check and see whether this black-box approach can resolve the problem more directly.

The contents in Section 3.1 are very interesting, particularly about the transformation in the domain set. Interestingly, I recently saw a paper for non-stochastic control (https://arxiv.org/abs/2206.09257) also employing a similar spirit to ensure the feasibility of the output decision and also keeping the exp-concavity. As the arXiv paper is posted very recently, so the paper and this submission should be deemed as concurrent work, but a discussion is believed necessary now.

In Theorem 3.1, the authors are encouraged to highlight the dependence on the exp-concavity coefficient $\alpha$, which is also a standard convention in OCO literature -- recall that the static regret is O(d/\alpha \cdot \log T).

Other concerns include some statements of the prior works, latex usages:
1) P3. Section 1.2 “Dynamic Pricing” should follow with a “dot”; this non-stationary dynamic pricing problem (especially the agnostic setting) is quite related to the area of non-stationary bandits, particularly, non-stationary linear bandits and generalized linear bandits (https://arxiv.org/abs/1903.01461; https://arxiv.org/abs/2103.05324). See discussions in (Luo et al., 2021; page 5, Bandit algorithms).

2) P4. On dynamic regret paragraph. One paper between (Zhang et al., 2018) and (Baby & Wang, 2021) should be mentioned --- when the loss functions are convex and gradient Lipschitz, problem-dependent universal dynamic regret can be achieved (https://dl.acm.org/doi/abs/10.5555/3495724.3496773; same regret but with improved gradient query complexity is achieved in https://arxiv.org/abs/2112.14368). It might be also helpful to the readers if the authors are willing to discuss recent efforts for other OCO settings and online control problems (https://arxiv.org/abs/2102.03758; https://arxiv.org/abs/2206.09257) and the bandit setting (https://arxiv.org/abs/1907.12340).
3) The appendix requires some careful proofreading, as there are many typos and inconsistency between appendix and main text. E.g.,
- Line 2 in Page 15, “Equation (9)” --> “Eq. (9)”
- the whole proof in Appendix B.3, the latex usage of the inner product is completely wrong. Please do not use “<” and “>”, the correct usage should be “\langle” and “\rangle”, see guideline https://www.atqed.com/latex-inner-product
- etc
4) The usage of reference (citep vs citet) is not used appropriately, see the guideline in JMLR (https://www.jmlr.org/format/format.html#reference-section). A basic rule is like “In general, you shouldn't have parenthetical statements embedded in parenthetical statements. Therefore, citations within parenthetical statements should not be embedded in parentheses.” So, many places used in this paper (I guess you use \citet or \cite) should be replaced by \citep

Asides from above comments, some minor typos include

5) P2. In “Proper Learning” paragraph, Line 5, a space is omitted before “In this work”.

6) P4, Table 1, is it possible to add two rows for lower bounds of static/dynamic regret, respectively? Or simply the one for dynamic regret.

7) P5. Eq. (3), better use non-italicized font for “Reg”, for example, “$\text{Reg}$”; also elsewhere

8) P8, Line 2, “the following Lemma” --> “the following lemma”

To summarize, I believe the paper studies an interesting problem and contributes nice insight to the community. The overall paper writing is of high quality (though some proofreadings are required), so I recommend the acceptance.

---

> ### Author Response · Authors · 2022-08-16
> **Response to Reviewer 1vaT**
>
> We thank the reviewer for your detailed and valuable comments!
>
> (Section 3.4): As for your concern regarding Section 3.4, we have re-organized this subsection and make it clearer. In specific, ADA-ILTCB$^+$ is a non-stationary contextual bandit algorithm that can achieve a $\tilde{O}(\sqrt{KT\log|\Pi|} + (K\log|\Pi|)^{\frac13}\Delta_T^{\frac13}T^{\frac23})$ *dynamic regret* (where $\Delta_T$ is the *distributional total variation* that can be upper bounded by $O(C_T)$). At each round $t$, it takes a context $x_t$ as an input, chooses a time-variant policy $\pi$ (which maps) from the policy set $\Pi$ and take an action $a_t=\pi_t(x_t)$ according to the policy's advice.
>
> (Comparison to [1]): We thank the reviewer for pointing this work. As far as we understand, the techniques in [1] do not exploit the curvature of the losses arising in our dynamic pricing setting to yield the fast dynamic regret rate of $O(T^{1/3})$. However, it is an interesting direction to derive similar black-box reductions that can also exploit the curvature of the losses in the more general non-stationary bandit setting.
>
> (Comparison to [2]): We would like to mention that in terms of timeline, we are actually earlier. The authors of [2] were aware of our work since our paper was initially submitted to ICML.  Initially they were thinking they could apply our results to the control problem but ended up needing to develop something new. In particular, they use a reduction from LQR problem to online (mini-batch) linear regression problem. The (mini-batch) linear regression problem is a bit different than our setting in that they don't have gradient direction before making predictions. This forced them to use the constant hessian property of linear regression losses. As such it seems that their techniques do not generalize beyond linear regression. Hence the results in this paper are not directly implied by the results of the LQR work. The authors of [2] could not, however, cite us because the manuscript is not available publicly.
>
> (Dependence on exp-concavity): We have highlighted the dependence on exp-concavity factor in Theorem 3.1 itself. As of now the dependence is displayed in the Equation at the bottom of page 8. The exp-concavity factor is $\gamma$ in that equation.
>
> (Dynamic Pricing Literature Review): We have extended our literature review in Paragraph "Dynamic Pricing" in Section 1.2 by including discussions on non-stationary contextual bandits, as you have suggested. This is closely related to our analysis in Section 3.4.
>
> (Typos, latex usages and minor issues): We thank the reviewer for these detailed suggestions. We have fixed these issues and converted all "Equation ($n$)" to "Eq. ($n$)" for $n\in\mathbb{Z}$.
>
> Reference:
>
> [1] Non-stationary Reinforcement Learning without Prior Knowledge: An Optimal Black-box Approach, Chen-Yu Wei and Haipeng Luo, COLT 2021
>
> [2] Optimal Dynamic Regret in LQR Control, Dheeraj Baby and Yu-Xiang Wang.

---

### Review · Reviewer_MWcH · 2022-07-20

**Summary Of Contributions:**

This paper discusses an application of online learning to price selection. It describes a setting in which buyer valuations are random variables with means $\langle x_t, \theta^\star\rangle$ for $x_t$ a context vector describing the product to be bought and $\theta^\star$ some preference parameter. The seller sees $x_t$, sets a price $v$ and gains reward $v$ if the $v$ is lower than the buyers valuation of the item. Assuming that the price is a greedy choice given some estimate $\theta_t$ for $\theta^\star$, then under appropriate conditions on the noise of the buyer valuation, it is possible to define a loss function $\ell_t(\theta)$ of in terms of whether a purchase was made whose expectation is equal to the expected reward of the greedy strategy using this $\theta_t$. Moreover, this loss is exp-concave. Thus, we could an exp-concave online algorithm like ONS to obtain logarithmic regret. It is my understanding that this much is the background setup, mostly from Xu & Wang 2021.

Now, the authors wish to obtain low dynamic regret by leveraging recent advance in exp-concave dynamic regret by Baby & Wang 2021 to obtain low dynamic regret when the preference vector $\theta^\star$ is slowly varying with path length $C_T$ without knowing $C_T$ in advance. Further, to enforce some kind of realism or safety on the described prices, the estimates $\theta_t$ must lie in some specified convex set. Baby & Wang's algorithm only works when the set is a $L_\infty$ ball. To expend beyond the ball, the authors introduce a reduction from optimization over any domain contained in the $L_\infty$ ball to optimization over the $L_\infty$ ball that takes inspiration from the reduction of Cutkosky & Orabona 2018. The original reduction does not allow for logarithmic regret on exp-concave losses because it generates a non-exp-concave surrogate loss. The authors proposal leverages the fact that the algorithm gets to see the feature $x_t$ before making a decision, and that the final gradient will be proportional to $x_t$. This enables construction of a more refined surrogate that preserves exp-concavity. As a result, the authors are able to relatively quickly port over the results for exp-concave losses to achieve a dynamic regret of $\tilde O(d^3 T^{1/3} C_T^{2/3})$.

An extension is provided that does not require as much information about the noise distribution on the buyer valuations.


**Broader Impact Concerns:**

I have no ethical concerns.

**Requested Changes:**

The nomenclature issue with "properness" may not be critical since the technical content makes sense on its own, but it is very confusing and so I strongly recommend that it be fixed.

**Strengths And Weaknesses:**

The paper is clearly written and the problem is interesting and well motivated.
The technical technical approach seems reasonable and is well explained. It provides a nice extension of the result of Cutkosky & Orabona to exp-concave linear supervised learning problems.

I do not believe the algorithm can be considered "proper" as it is normally understood in online optimization. The choice for $\theta_t$ depends on the revealed feature $x_t$ through use of the $x_t$-dependent projection to the constraint set, while the comparator does not. So, I think this is in fact a standard example of an improper algorithm for linear learning tasks (e.g. see https://arxiv.org/abs/1803.09349 for an example of this type of improperness).
This is not serious concern - the requirement to have $\theta_t$ inside a specific constraint set is still an unsolved problem even with an improper choice for $\theta_t$, but a different phrase should be used.

Some minor/typo level issues:

The set is "closed", not "close" - this typo/small mistake appears several times.

Please state that $f = F’$ in your definitions. I do not think $f$ is actually defined anywhere.

**EDIT**: the original review posting contained a few concerns that I realized were not actually issues.

---

> ### Author Response · Authors · 2022-08-16
> **Response to Reviewer MWcH**
>
>
> We thank the reviewer for your valuable and detailed comments!  We have fixed the typos and other minor issues according to your suggestions. In the following, we will make clarification on the notion of ``proper learning''.
>
> ("Proper learning" definition):  We thank the reviewer for bringing Foster et al. (2018) and their definition of "Proper Learning" to our attention.  It looks like they define "proper learning" to be a combination of two things:
>
>     (1) One chooses weight $w_t$ before observing $(x_t,y_t)$
>     (2) $w_t\in \mathcal{W}$ for a given convex set $\mathcal{W}$.
>
> This makes sense for them because this is the setting of standard OCO when applying to the problem of online logistic regression.  In standard OCO,  $w_t$ should not depend on $x_t$.
>
> Our setting is a bit different. We do not claim that we solved the proper dynamic regret minimization problem for general OCO. Instead, we focus on the contextual dynamic pricing problem in which the observation of $x_t$ at the time of prediction is quite natural. Of course, our results work for other linear prediction problems where $x_t$ is observed, but that remains a narrower scope comparing to the generic OCO problem.
>
> If we take a step back and stare at the two conditions in Foster et al. (2018).  (1) is really about the information set available to the learner; and (2) is about whether the learning algorithm is (or is not) restricted to output a hypothesis from $\mathcal{H}$. In our context, (1) and (2) are complementary considerations and it seems unnecessarily confusing  to couple them.  As a matter of fact, in the classical learning theory literature, "proper (or improper) learning" is really only about  (2), i.e. "the learning algorithm is (or is not) restricted to output a hypothesis from $\mathcal{H}$" (quoted directly from Daniely et al. (2013) ).
>
> For these reasons, we think it is appropriate for us to use the term "proper" in our problem.  To avoid confusion, we stated our definition in the *Proper Learning* paragraph in Sec 1 and discussed the difference from Foster et al. (2018) in the related work section.
>
> Reference:
>
>     Daniely et al. (2013): Daniely, Amit, Nati Linial, and Shai Shalev-Shwartz. "More data speeds up training time in learning halfspaces over sparse vectors." in NIPS 2013
>
>     Foster et al. (2018): Foster, Dylan J., Satyen Kale, Haipeng Luo, Mehryar Mohri, and Karthik Sridharan. "Logistic regression: The importance of being improper." in COLT 2018.

---

### Review · Reviewer_Br6s · 2022-08-10

**Summary Of Contributions:**

The paper studies a contextual dynamic pricing problem where the environment can change over time. They establish optimal dynamic regret rates scaling with a total variation quantity, with the main proof strategy being a reduction to online convex optimization.

**Broader Impact Concerns:**

The work is entirely theoretical so I don't foresee any broader impact concerns.

**Requested Changes:**

Please see above.

Minor typos:
* There should be a $J(x^T\theta_t)$ on the LHS of the display immediately before section 3.
* Display (12) is missing an absolute value sign.
* The last term in parentheses of both RHS's of display (17) should be squared.
* In the bottom of page 16, "rxp-concavity" is misspelled.
* The inequality next to $\|\beta^{(i)}-\beta^{(j)}\|_1$ at the bottom of page 17 should not be there.

**Strengths And Weaknesses:**

Overall, I think the result is interesting and novel, and the paper is also well written making clear the technical intuition and novelties over previous works. So I am generally in support of acceptance.

My specific questions are as follows:
1. I am curious if $C_T$ is an appropriate notion of non-stationarity in this setting. It seems like only changes in the valuations $x_t^T\theta_t^*$ w.r.t. the fixed sequence $\{x_t\}_t$ should affect the regret. Is the dependence on $C_T$ a consequence of reducing the problem to estimating $\theta_t^*$ or because the context sequence is assumed to be adversarial? Can regret bounds be given in terms of changes in the valuations?
2. I am also confused about Remark 3.9. The dependence of the regret bounds on the noise parameters is not shown and so it is not clear how the regret scales with them or whether the dependence is optimal.
3. I also think it should be made more clear what parameters of the noise distribution are required by the algorithm and whether these assumptions are plausible, as such matters are now hidden away in the lemmas.
4. The regret bounds in the noise agnostic setting do not seem to be tight and even vacuous for certain regimes of $T$ and $C_T$. I am hoping the authors could comment more on what is possible in this setting.
5. In terms of writing, I think Section 3.4 would benefit from a clear explanation of how the contextual dynamic pricing problem is reduced to contextual bandits, as the current writing is missing this.

---

> ### Author Response · Authors · 2022-08-16
> **Response to Reviewer Br6s**
>
> We thank the reviewer for your detailed and valuable comments!
>
> ($C_T$ as a notion of non-stationarity): We chose to capture the variation in terms of the change in the parameter $\theta_t^*$ instead of the valuations $x_t^T\theta_t^*$ because, the valuations can change abruptly across time. This can result in a linear path length wrt changes in valuations ($\sum_{t=2}^T |x_t^T \theta_t^* - x_{t-1}^T\theta_{t-1}^*|$) making it a not so appropriate metric to capture the degree of non-stationary. For example, consider the scenario where the contexts are drawn iid from a uniform distribution over $[-1,1]$, while $\theta_t^*= 1$ remains fixed for all $t$. In this case, while the path length wrt valuations is linear in $T$ with high probability, there is indeed a static policy that is optimal in hindsight.
>
> (Remark 3.9): The dependence of regret on the noise distribution lies in the coefficient $\frac{C_{down}}{C_{exp}}$, where $C_{down}$ and $C_{exp}$ (defined in Lemma 3.7 and 3.8 correspondingly) are dependent on the noise distribution (CDF $F(\cdot)$ and PDF $f(\cdot)$). We do not know if this dependence is optimal, as we assume that the noise distribution is fixed over time and we may treat every parameter on the noise as a constant.
>
> Our analysis in Lemma 3.7 and Lemma 3.8 adopts the same approach as Xu \& Wang (2021), where they give an example of how the regret depends on the Gaussian noise standard deviation $\sigma$ (see their Appendix C.1). They show an exponential increase on the regret as $\sigma$ gets close to zero.
>
> (Required assumptions for algorithms and lemmas): Thanks for your suggestions! We have included the noise distribution $\mathbb{D}$ as an input of our Algorithm 2 (PDRP) and have stated the assumptions of noise knowledge in these related theorems/lemmas in our revised paper. In fact, the PDRP algorithm requires *all* information of the noise distribution, while the Algorithm 3 (D2-ADA) requires *no* information of it.
>
> (No tight bounds for noise-agnostic setting): We agree with the reviewer that the *dynamic* regret upper bound here is not tight, and intuitively it should be sub-optimal with respect to $T$. However, this is reasonable since the optimal *static* regret bound is still unclear so far. We have made major revision on Section 3.4, and hope the revised version would make our paper clearer.
>
> (Reduction to contextual bandits in Sec 3.4): We have added a description of the reduction from our contextual pricing problem to a contextual bandit problem setting. Also, we have made major revision on Section 3.4 to comply with these changes.
>
> Besides, we have corrected all typos as you suggested. Thanks for pointing them out!

---

### Comment · Reviewer_MWcH · 2022-07-21
**review updated**

Please note that my review has been updated since its initial posting.

---

### Author Response · Authors · 2022-08-16
**To All Reviewers**

For all reviewers:

We thank you all for your valuable comments! Please kindly find our individual response below your reviews. We have also revised our submission according to your suggestions, and highlight those changes in purple.

Yours,

Authors of Paper216

---

### Decision · Action_Editors · 2022-09-29

**Recommendation:** Accept with minor revision

**Comment:**

All reviewers were positive about the paper and recommended acceptance, and I also share their opinion. While their corrections and suggestions have mostly been incorporated in the manuscript, I have found a number minor issues which should be addressed/corrected in the final version.

- Abstract: Please define notation ($T$, $C_T$, $d$) used in the abstract. It would be great to mention that you consider a linear pricing model for the users, and for online learning with exp-concave losses, the attention is restricted to loss functions coming from a generalized linear model. Consider mentioning explicitly that the bound may not be optimal in $d$ and also the $\log T$ dependence (hence the dependence on $T$ is only "almost" optimal). The assumption in the paper on the noise is more restrictive than log-concave (as you require that $1-F$ is also log-concave). (Also, dynamic regret is measured relative to "the" optimal sequence of policies, not to an arbitrary one.)

- Section 3.4: Please significantly improve the presentation of this section. Make the statements of Theorem 3.10, Theorem 3.11, and Lemma 3.12 precise: For Theorem 3.10, follow what is given in the appendix: for every algorithm there exists a problem such that... For Theorem 3.11 and the lemma, describe more precisely (quantify) what you mean by high probability. Mention discretization early on in the section, otherwise it is not clear how the reduction to bandits (with finitely many actions) can be done. Add a brief description of ADA-ILTCB+. What is ADA-ILTCB*? It would be good to have a short discussion about how the bandit reduction does not fully utilize the observations (estimating the reward for other prices) and how it leads to deteriorated rates compared to the previous results. The wording is awkward at several places, including "$N_t$'s are absolutely bounded by constants", "each policy takes into an $x_t$", "integer multiplying $\Delta$/$\gamma$", "constantly bounded", "the distribution $D$ are fully agnostic", "probabilistic distribution" (should actually be density function), "distributional switching", etc.

Minor comments/typos:

- p. 1, last line: i.e -> i.e.
- p. 2, around equation (1): define $w_{1:T}=(w_1,\ldots,w_T)$
- Please make sure citations are properly integrated into sentences, e.g., on p. 2, 2  lines below equation 1, it should be (Zinkevich, 2003), or on p. 4, in the "Dynamic Regret" paragraph, "by Zhao et al. (2020b)" seems better, or on p. 4, there are too many parentheses in (e.g. (Zhao et al. 2022...)). This has already been mentioned by Reviewer 1vaT.
- p. 4, last line of the "Dynamic Regret" paragraph: "arbitrary decision sets" should be "arbitrary bounded convex decision sets"
- in the next paragraph "algorithms falls".
- p. 4, paragraph above Section 2: It is not entirely clear what you mean by having a gradient direction before making the prediction.
- p. 4, l. -2: $\theta^*$ is not defined in the text.
- p. 5, Section 2.2: it would be good to mention here again that the sets ($\mathcal{D}_x$, etc.) are convex.
- p. 5: Please discuss the assumptions that $F$ and $1-F$ are log-concave (in particular, it would be great if you could present some examples, like the sigmoid function).
- p. 5, paragraph about restrictions: It is not clear to me what you mean here by "without loss of generality". You introduced a restriction that "When we take an action by presenting a price $v_t$,
there always exists an $\theta_t$ such that $x^\top \theta_t = J^{-1}(v_t)$." and the next statement follows from this, but why can you do this without loss of generality?
- p. 6, Consistency: "we requires".
- p. 7: "losses $\ell_t$ exhibits".
- p. 7: Theorem 3.1/Remark 3.2: Please discuss the corresponding lower bound (especially since the remark talks about "optimal" adaptivity to $C_T$, hence an argument substantiating optimality would be welcome).
- p. 8: Above Lemma 3.3: $\hat{l}_t$ should be $\hat{\ell}_t$.
- p. 9, Theorem 3.6: $C_{down}$ and $C_{exp}$ are only defined in the subsequent lemmas. Please make sure they are defined before use. Compared to Theorem 3.1, a factor of 2 seems to be missing in the definition of $\beta$ (corresponding to the $\alpha/2$ term.
- p. 10, after Lemma 3.8: should be "Lemma 7 of Xu & Wang (2021)". Last line of the proof: "Theorem"->theorem.
- p. 10, Remark 3.9: You could add here your response to Reviewer Br6s that Xu and Wang "show an exponential increase on the regret as $\sigma$ gets close to zero."
- p. 10, Sec 3.3: "have construct".
- p. 11, line 4: delete "While".


**Audience:**

The proposed model, algorithms and analysis are of interest to the online-learning community. Probably technically the most interesting contribution is the analysis of the dynamic regret for proper online learning with exp-concave losses (coming from a family of generalized linear models) over arbitrary bounded convex domains, combining and extending previous results from the literature (see the summary of contributions by Reviewer MWcH).

**Claims And Evidence:**

The reviewers found that the claims of the paper are correct.